# Aspirin modulates production of pro-inflammatory and pro-resolving mediators in endothelial cells

Kara M. Rood[1]\*, Niharika Patel[2], Ivana M. DeVengencie[2], John P. Quinn[2], Kymberly M. Gowdy[3,4], Maged M. Costantine[1], Douglas A. Kniss[1,2,5,6]

1 Division of Maternal-Fetal Medicine, Department of Obstetrics and Gynecology, College of Medicine and Wexner Medical Center, The Ohio State University, Columbus, Ohio, United States of America, 2 Division of Maternal-Fetal Medicine, Department of Obstetrics and Gynecology, Laboratory of Perinatal Research, College of Medicine and Wexner Medical Center, The Ohio State University, Columbus, Ohio, United States of America, 3 Division of Pulmonary, Critical Care and Sleep Medicine, Department of Internal Medicine, College of Medicine and Wexner Medical Center, Columbus, Ohio, United States of America, 4 Dorothy Davis Heart and Lung Institute, College of Medicine and Wexner Medical Center, The Ohio State University, Columbus, Ohio, United States of America, 5 Department of Biomedical Engineering, College of Engineering, Fontana Labs, The Ohio State University, Columbus, Ohio, United States of America, 6 Infectious Disease Institute, The Ohio State University, Columbus, Ohio, United States of America

\* kara.rood@osumc.edu

**Data Availability Statement:** All relevant data are within the manuscript and its Supporting Information files.

## Abstract

Endothelial cells synthesize biochemical signals to coordinate a response to insults, resolve inflammation and restore barrier integrity. Vascular cells release a variety of vasoactive bio-active lipid metabolites during the inflammatory response and produce pro-resolving mediators (e.g., Lipoxin $A_4$, $LXA_4$) in cooperation with leukocytes and platelets to bring a halt to inflammation. Aspirin, used in a variety of cardiovascular and pro-thrombotic disorders (e.g., atherosclerosis, angina, preeclampsia), potently inhibits proinflammatory eicosanoid formation. Moreover, aspirin stimulates the synthesis of pro-resolving lipid mediators (SPM), so-called *A*spirin-*T*riggered *L*ipoxins (ATL). We demonstrate that cytokines stimulated a time- and dose-dependent increase in $PGI_2$ (6-keto$PGF_{1\alpha}$) and $PGE_2$ formation that is blocked by aspirin. Eicosanoid production was caused by cytokine-induced expression of cyclooxygenase-2 (COX-2). We also detected increased production of pro-resolving $LXA_4$ in cytokine-stimulated endothelial cells. The R-enantiomer of $LXA_4$, 15-epi-$LXA_4$, was enhanced by aspirin, but only in the presence of cytokine challenge, indicating dependence on COX-2 expression. In contrast to previous reports, we detected arachidonate 5-lipoxygenase (ALOX5) mRNA expression and its cognate protein (5-lipoxygenase, 5-LOX), suggesting that endothelial cells possess the enzymatic machinery necessary to synthesize both pro-inflammatory and pro-resolving lipid mediators independent of added leukocytes or platelets. Finally, we observed that, endothelial cells produced $LTB_4$ in the absence of leukocytes. These results indicate that endothelial cells produce both pro-inflammatory and pro-resolving lipid mediators in the absence of other cell types and aspirin exerts pleiotropic actions influencing both COX and LOX pathways.

**Funding:** Award, KMR Grant No. UL1TR002733 Center for Clinical and Translational Sciences, Ohio State University/NIH NCATS ccts.osu.edu; nih.gov Sponsor played no role in the study design, data collection and analysis, decision to publish, or preparation of the manuscript.

**Competing interests:** The authors have declared that no competing interests exist.

## Introduction

During local or systemic inflammation, vascular endothelial cells serve as both target for and producer of mediators, including cytokines, chemokines and bioactive lipids derived from polyunsaturated fatty acids (PUFA) [1]. Historically, acute inflammation was thought to be driven by mediators and the termination of inflammation was thought to be a passive process in which the injurious signals were simply inactivated or diluted [2]. The contemporary paradigm, however, suggests that resolution of inflammation is an *active process*, orchestrated, in large part, by a group of bioactive lipids known as Specialized Pro-resolving Mediators (SPMs) [3] that are synthesized locally shortly after the onset of inflammation. Failure to produce SPMs in a timely manner leads to persistence of infection or injury, sustained leukocyte infiltration and release of reactive oxygen species (ROS) and other molecules that can damage tissues and convert an acute inflammatory response into one that can provoke the onset of chronic disease such as cardiovascular diseases [4–7].

Lipoxins, a family of SPMs are formed by transcellular biosynthesis as the 15S-hydroperoxyeicosatetraenoic acid (15S-HpETE) intermediate generated in endothelial cells by arachidonate 15-lipoxygenase-1 (ALOX15, 15-LOX-1) that is converted by neutrophil arachidonate 5-lipoxygenase (ALOX5, 5-LOX) to synthesize positional isomers $LXA_4$ or $LXB_4$ [8, 9]. An alternative pathway involves the cooperation between neutrophil ALOX5 and platelet ALOX12 (12-LOX) to manufacture $LXA_4$ and $LXB_4$ [10]. A third pathway for lipoxin biosynthesis is based on the mechanism of action of the classical non-steroidal anti-inflammatory drug (NSAID), aspirin [11]. In this scheme, the acetyl moiety from aspirin covalently acetylates serine[516] within the active site of endothelial cell cyclooxygenase-2 (COX-2) to generate 15R-hydroxyeicosatetraenoic acid (15R-HETE) that is used by leukocyte 5-LOX to catalyze the formation of 15-epi-$LXA_4$ or 15-epi-$LXB_4$, stereoisomers of $LXA_4$ and $LXB_4$, respectively [12, 13]. These so-called aspirin-triggered lipoxins (ATLs) are highly potent and, along with their 15 (S)-related stereoisomers, halt influx of neutrophils into sites of inflammation, restore barrier function in the endothelium, activate macrophages to undergo phagocytosis of apoptotic neutrophils and cell debris (efferocytosis) and foster restoration of tissue homeostasis [14–16]. Importantly, aspirin also acetylates and irreversibly inactivates COX-1 which is expressed in endothelial cells, begging the question of the relative contribution of both COX isoforms to the formation of proinflammatory and anti-inflammatory/pro-resolving lipids and the mechanism of aspirin action in modulating inflammation at the endothelial-leukocyte interface [17–21].

To date, most investigations of SPM formation have focused on the transcellular synthesis of lipoxins and other lipid mediators, requiring at least two cell types to carry out the complete reaction sequence [22, 23]. Very few studies have addressed the potential of endothelial cells to produce pro-resolving lipids in the absence of leukocytes or platelets. In the present study, we tested the hypothesis that human endothelial cells express the enzyme machinery necessary to biosynthesize lipoxins involved in the resolution of inflammation and could do so independently of leukocytes (neutrophils or macrophages) or platelets. We demonstrate that proinflammatory cytokines (interleukin −1β, IL−1β and tumor necrosis factor−α, TNFα) induced COX-2 and 5-LOX and stimulated proinflammatory eicosanoid production, including leukotriene $B_4$ ($LTB_4$), and stimulated the formation of pro-resolving $LXA_4$, in addition to $PGI_2$ (measured as $6ketoPGF_{1\alpha}$) and $PGE_2$. Furthermore, aspirin stimulated the synthesis of 15-epi-$LXA_4$ in endothelial cells without the assistance of neutrophils.

## Materials and methods

### Materials

Human umbilical vein endothelial cells (HUVEC) were purchased from ATCC (PCS-100-013) and Lonza Biosciences (Basel, CH, pooled donors, catalog no. C2519A). Culture media and

supplements (EBM-2™ medium and EGM-2 Bullet kit) were purchased from Lonza (Basel, CH). Trypsin-EDTA solution (0.25%-0.9 mM) was purchased from ThermoFisher Scientific (Waltham, MA). Cytokines (human recombinant interleukin−1β, IL−1β and tumor necrosis factor−α, TNFα) were purchased from R&D Systems (Minneapolis, MN). Primary antibodies were supplied by Abcam (Waltham, MA), Invitrogen/ThermoFisher Scientific (Waltham, MA), Sigma/Aldrich (St. Louis, MO), ProteinTech (Rosemount, IL) and Cayman Chemical Co. (Ann Arbor, MI) (S1 Table). Secondary antibodies (goat anti-rabbit IgG-HRP, goat anti-mouse IgG-HRP) were purchased from Invitrogen/ThermoFisher Scientific. Enzymes, buffers and fetal bovine serum were purchased from GIBCO (ThermoFisher). Aspirin and salicylic acid were obtained from Sigma/Aldrich. Arachidonic acid, nordihydroguariatic acid (NDGA), Zileuton and MK-886 were purchased from Cayman Chemical Co. (Ann Arbor, MI). All other chemicals were tissue culture or reagent grade Sigma-Aldrich.

## Cell culture

Primary cultures of human umbilical endothelial cells (HUVEC) were grown in EBM-2 basal medium with EGM-2 supplements. Cells were initially seeded into 75 cm$^2$ tissue culture flasks in complete EGM-2 medium and grown to ~90% confluence. Medium was changed every two days. For subculture, cells were dislodged with 0.25% trypsin-EDTA, counted using a Scepter™ handheld automated cell counter (Millipore, Burlington, MA) and seeded into appropriate tissue culture vessels (48-well plates, 5 x 10$^4$/well; 60 mm dishes, 2 x 10$^5$/dish; 12-well plates, 1 x 10$^5$/well). HUVEC were cultured for no more than seven passages before replacing with fresh cells.

## Protein extraction and western blotting

HUVEC seeded into 60 mm tissue culture dishes (2 x 10$^5$/dish) in complete medium were grown to confluence. Cells were rinsed once with Hank's Balanced Salt Solution (HBSS, pH 7.2) and then incubated in test substances in EGM-2 complete medium. Total protein was extracted with RIPA buffer (150 mM NaCl, 50 mM Tris, pH 7.4, 1% NP-40, 0.1% sodium dodecyl sulfate, SDS, 0.5% deoxycholate, DOC, 1 mM ethylenediamine tetraacetic acid, EDTA) containing protease inhibitor cocktail (Sigma-Aldrich, St. Louis, MO) and 1 mM phenylmethylsulfonylfluoride, PMSF) on ice for 30 min, followed by centrifugation at 15,000 x g for 15 min at 4˚C. Protein concentration was determined using detergent-compatible Biorad DC® reagent (Biorad, Hercules, CA) with bovine serum albumin (BSA, Sigma) as standard and 30 μg/lane were fractionated in 4–20% sodium dodecyl sulfate-polyacrylamide gels (SDS-PAGE, NuPage™ gels, Invitrogen, Waltham, MA). Proteins were transferred to nitrocellulose membranes using the semi-dry iBLOT™ device (Invitrogen), followed by blocking in Tris-buffered saline-0.2% Tween-20 (TBST) containing 5% nonfat dry milk (BLOTTO). Membranes were probed with primary antibodies (S1 Table) in TBST/2% BSA or 5% non-fat dry milk for 60 min at room temperature. Blots were washed three times with TBST and then incubated with secondary antibodies: goat anti-rabbit IgG (H+L)-horseradish peroxidase (HRP, 1:2000–1:10,000 in TBST/2% BSA) for 60 min at room temperature. After thorough washing of blots with BLOTTO and TBST, immuno-reactive proteins were visualized using chemiluminescence (Clarity® ECL reagent, Biorad, Hercules, CA) and digital imaging with the Biorad ChemiDoc-MP™ system. Densitometry data were computed using Biorad ImageLab™ 6.1 or ImageJ (ImageJ.nih.gov) software.

## RNA extraction and qRT-PCR

Cells were seeded into 6-well plates (2 x 10$^5$/dish) in complete medium and grown to confluence. To initiate experiments, cells with rinsed once with HBSS and then incubated with test

substances in complete medium for the times indicated in figure legends. At the conclusion of experiments, cells were rinsed once with cold PBS and then scraped into 1.5-ml microfuge tubes, pelleted in a refrigerated microfuge and frozen at -80C. Total RNA was extracted using Trizol® (ThermoFisher) and chloroform according to manufacturer's specifications. After precipitating with cold 95% ethanol, RNA was quantified using a NanoDrop™ spectrophotometer ($A_{260/280}$) (ThermoFisher Scientific). To generate cDNA, 1 μg/tube of total RNA was reversed transcribed with LunaScript® RT SuperMix (New England Biolabs, Ipswich, MA). Polymerase chain reactions were carried out with Taqman® Fast Universal PCR Master Mix Taqman® Gene Expression Assay primer sets (ThermoFisher Scientific) (S2 Table).

## ELISA

Cells seeded into 48-well plates (5 x $10^4$/well) or 12-well plates (1 x $10^5$/well) in complete medium and grown to confluence. Cultures were rinsed with HBSS and shifted to M199+0.5% charcoal-stripped FBS/5% bovine endothelial cell growth supplement (ECGS, ThermoFisher Scientific) for experiments. Culture media were collected, cells were solubilized with 1N NaOH and samples were frozen -80˚C. Specific ELISAs for 6-ketoprostaglandin $F_{1\alpha}$ (6-ketoPGF$_{1\alpha}$), prostaglandin $E_2$ (PGE$_2$), Lipoxin $A_4$ (LXA$_4$), 15-epi-Lipoxin $A_4$ (15-epi-LXA$_4$) and Leukotriene $B_4$ (LTB$_4$) (Cayman Chemical, Ann Arbor, MI) were used to measure analytes following incubation with test substances. Total cell protein was measured in each well and data were expressed as ng of analyte/mg protein. All measurements were made using a Perkin Elmer Victor V3™ multichannel microplate reader. To control for possible endogenous lipid mediators in the culture media, each analyte was measured in culture media (M199 +0.5% FCS/5% ECGS) that was not incubated with cells.

## Statistics

ELISA data were expressed as mean±SEM (ng of analyte produced/mg protein) of n = 6-12/ condition in 3–4 replicate experiments. Data for western blotting (n = 3–4 biological replicates in 2–3 experiments) and qRT-PCR (n = 4–6 biological replicates in 2–3 experiments) experiments were expressed as ratio of target/GAPDH (qRT-PCR, $2^{-\Delta\Delta Ct}$ method for qRT-PCR [24] or β–actin (western blot only) by densitometry using the ChemiDoc-MP™ Imaging System and *Image Lab*™ software (Biorad, Hercules, CA) and were run using biological replicates. Differences among treatments were determined using one-way analysis of variance (ANOVA) followed by Dunnett's or Tukey's multiple comparisons testing (p<0.05 considered significant, unless otherwise specified). Where appropriate, Student's t-test for differences was employed to compare control and experimentally treated samples. All datasets were tested for homogeneity of variance (i.e., normal distribution) prior to statistical analysis. All data were plotted and analyzed using GraphPad Prism™ 9.0 (v9.3.1) software (San Diego, CA).

## Results

### Cytokines stimulate pro-inflammatory, vasoactive eicosanoid production in endothelial cells

We compared the ability of IL−1β and TNFα to stimulate eicosanoid production in HUVECs. Cells were challenged with recombinant human IL−1β or TNFα and media collected at intervals from 0.25 to 24 hr for ELISA. Fig 1A illustrates the kinetics of PGI$_2$ (measured as 6-ketoPGF$_{1\alpha}$) and PGE$_2$ synthesis. After 4–6 hr, cells produced 6-ketoPGF$_{1\alpha}$ and PGE$_2$, peaking at 12 hr, and remaining elevated 24 hr post exposure. Differences between cytokine-treated and control cells became apparent 12 hr after IL−1β and TNFα stimulation, respectively. The

maximal level of 6-ketoPGF$_{1\alpha}$ was nearly 10-fold greater than PGE$_2$ for both IL−1β and TNFα. IL−1β at 2 ng/ml was approximately 4-fold more efficacious in stimulating 6-ketoPGF$_{1\alpha}$ synthesis than TNFα at 20 ng/ml. Similar efficacy for IL−1β compared to TNFα was seen for PGE$_2$ (7.5-fold, IL−1β vs. 3-fold, TNFα).

In the next experiment, we compared the dose-responses for IL−1β- and TNFα- stimulated eicosanoid, 6-ketoPGF$_{1\alpha}$ and PGE$_2$, synthesis. HUVECs were challenged 24 hr with IL−1β (0–20 ng/ml) or TNFα (0–50 ng/ml) and media were assayed 6-ketoPGF$_{1\alpha}$ and PGE$_2$ by ELISA. Fig 1B shows the dose-response curves for IL−1β (0.01–20 ng/ml) and TNFα (0.1–50 ng/ml). The EC$_{50}$ values for revealed that IL−1β was far more potent than TNFα for the two analytes (IL−1β: 6-ketoPGF$_{1\alpha}$− 0.1272 ng/ml, PGE$_2$−0.2894 ng/ml; TNFα: 6-ketoPGF$_{1\alpha}$− 39.6 ng/ml,

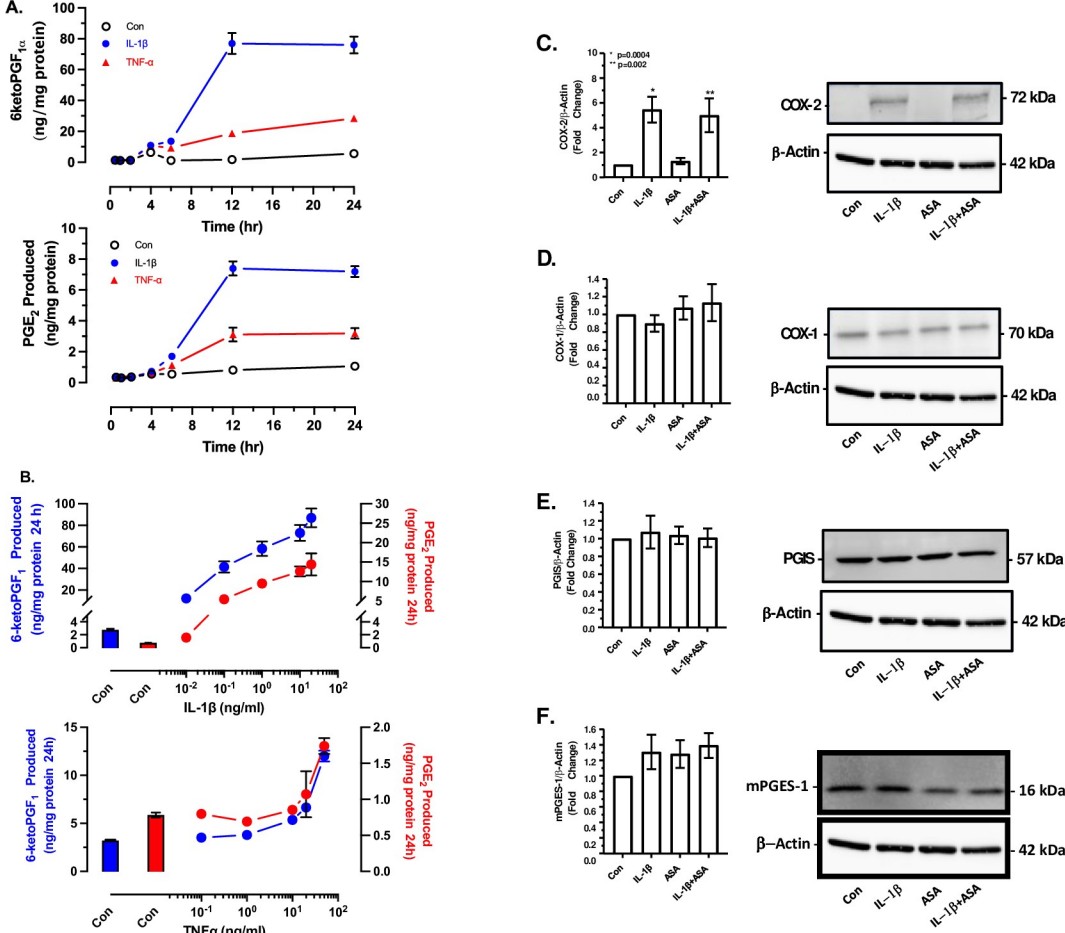

**Fig 1. Cytokines stimulate eicosanoid production in endothelial cells.** (A) **Kinetics:** HUVEC were stimulated with human recombinant IL−1β (2 ng/ml, blue circles) or TNFα (20 ng/ml, red triangles) for 0.25–24 hr in DMEM+0.5% FBS and assayed for 6-ketoPGF$_{1\alpha}$ (upper panel) or PGE$_2$ (lower panel). Control cells were incubated with vehicle (PBS/0.1% BSA, open circles). (B) **Dose-response:** HUVEC were stimulated with IL−1β (0.01–10 ng/ml) or TNFα (0.1–50 ng/ml) for 24 hr in DMEM+0.5% FBS and assayed for 6-ketoPGF$_{1\alpha}$ (stable metabolite of PGI$_2$) and PGE$_2$. Control cells were incubated with vehicle (PBS/0.1% BSA, 6-ketoPGF$_{1\alpha}$, blue bar; PGE$_2$, red bar). Data are expressed as mean±SEM (n = 6). Experiment was replicated three times. (A)*p<0.01; **p<0.0001. (B, upper panel) *p<0.0128; **p<0.0001; (B, lower panel) *p<0.0004; **p<0.0001. (C) Western blot: HUVEC extracts from vehicle or IL−1β (2 ng/ml)-stimulated cells were probed with COX-1, COX-2, PGIS or mPGES-1 antibodies. Blots were re-probed with glyceraldehyde 3-phosphate dehydrogenase (GAPDH) as a loading control. Immunoreactive proteins were visualized by chemiluminescence and the ratio of target:β−actin was computed by densitometry (D). The data are presented as the mean±SEM and represent of 3 biological replicates and experiment was conducted twice. The data were analyzed by one-way analysis of variance followed by Dunnett's test for multiple differences (p values are shown and a minimum p value of <0.05 was considered significant).

$PGE_2$–32.62 ng/ml). These data suggest that low to sub- nanogram concentrations of both proinflammatory cytokines were highly effective at stimulating further inflammatory mediator synthesis. As expected, increased synthesis of $PGI_2$ and $PGE_2$ was accompanied by a concomitant induction of cyclooxygenase-2 (COX-2), the inducible isoform of $PGH_2$ synthase. In contrast, COX-1, the constitutively expressed isoform of $PGH_2$ synthase, prostacyclin synthase (PGIS) and microsomal prostaglandin $E_2$ synthase-1 (mPGES-1) were unchanged assayed by western blotting and quantified by densitometry (Fig 1C–1F).

Given that IL–1β was more potent and efficacious than TNFα in stimulating eicosanoid synthesis in endothelial cells, we focused the remainder of the studies on inflammation driven by this cytokine. HUVEC were grown as described above and challenged with various concentrations of IL–1β (0.01–10 ng/ml) for 24 hr and total RNA was extracted for qRT-PCR (Fig 2). IL–1β did not alter the expression of COX-1, the constitutive isoform of the rate-limiting enzyme in prostaglandin production. In contrast, IL–1β stimulated a robust increase in COX-2 expression in a dose-dependent manner (Fig 2A). PGIS and mPGES-1 transcripts encoding the enzymes that produce $PGI_2$ and $PGE_2$, respectively were also up-regulated by IL–1β, albeit to a lesser extent than COX-2 (Fig 2B and 2C).

## IL–1β stimulates $LTB_4$ synthesis in endothelial cells independent of leukocytes: Modulation by aspirin

Historically, endothelial cells were thought to produce chemoattractant lipoxygenase products, e.g., Leukotriene $B_4$ ($LTB_4$), via transcellular biosynthesis only when co-incubated with polymorphonuclear leukocytes (neutrophils) [25]. For example, endothelial cells stimulated with calcium ionophore (A23187) failed to produce $LTB_4$, but when incubated with $LTA_4$ and A23187 produced Cysteinyl LTs (CysLT) [26]. Endothelial cell-neutrophil co-incubations resulted in $LTB_4$ production. In contrast, porcine endometrial endothelial cells produced $LTB_4$ and CysLTs ($LTC_4$, $LTD_4$ and $LTE_4$) in the absence of leukocytes [27]. To clarify this apparent discrepancy, we treated HUVEC with IL–1β (2 ng/ml) for 24 hr and noted a significant increase in $LTB_4$ production in the absence of leukocytes (Fig 3A). We detected a 1.5–3.0-fold increase (A, * p = 0.0036, ** p<0.0001) in $LTB_4$ formation in cells treated with IL–1β. To demonstrate that $LTB_4$ synthesis was dependent on LOX activity, IL–1β- or vehicle-treated cells were preincubated with NDGA, a broad-spectrum lipoxygenase inhibitor, and then challenged with A23187 for 30 min. NDGA completely blocked $LTB_4$ synthesis in HUVEC in both

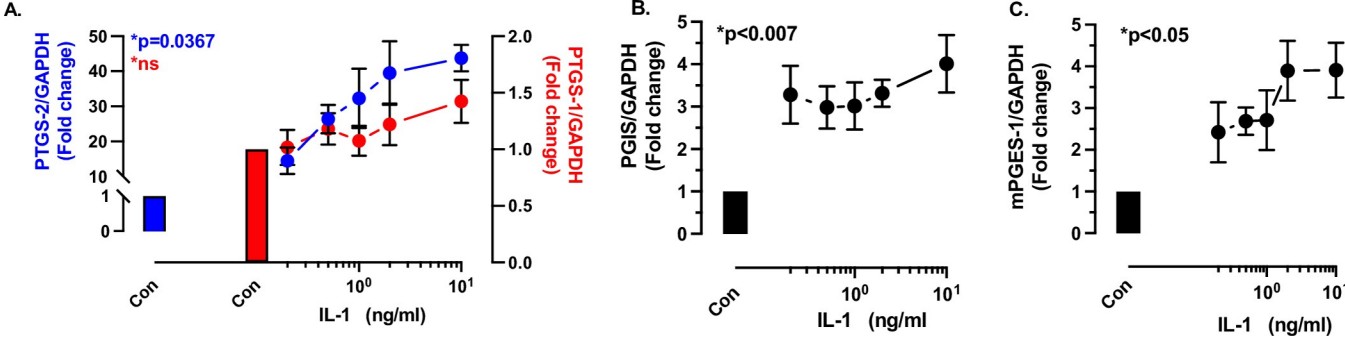

**Fig 2. IL–1β stimulates mRNAs encoding eicosanoids synthesizing enzymes in endothelial cells.** HUVEC were stimulated with various concentrations of human recombinant IL–1β (0.2–10 ng/ml) for 24 hr and total RNA was extracted and processed for qRT-PCR using Taqman™ chemistry. Control cells were incubated with vehicle alone (PBS/0.1% BSA). Primer sets for COX-1, COX-2, PGIS and mPGES-1 were used along with the housekeeping gene GAPDH. The $2^{-\Delta\Delta Ct}$ method was used to compute relative transcript expression levels. (A) COX-1, COX-2; (B) PGIS; (C) mPGES-1. The data are expressed as mean±SEM of 3 biological replicates and the experiment was conducted twice. (A) One-way ANOVA followed by Dunnett's multiple comparisons test was performed (*p<0.0367, blue; **ns, red). (B) two-tailed t-test, *p<0.007; (C) two-tailed t-test, p<0.05.

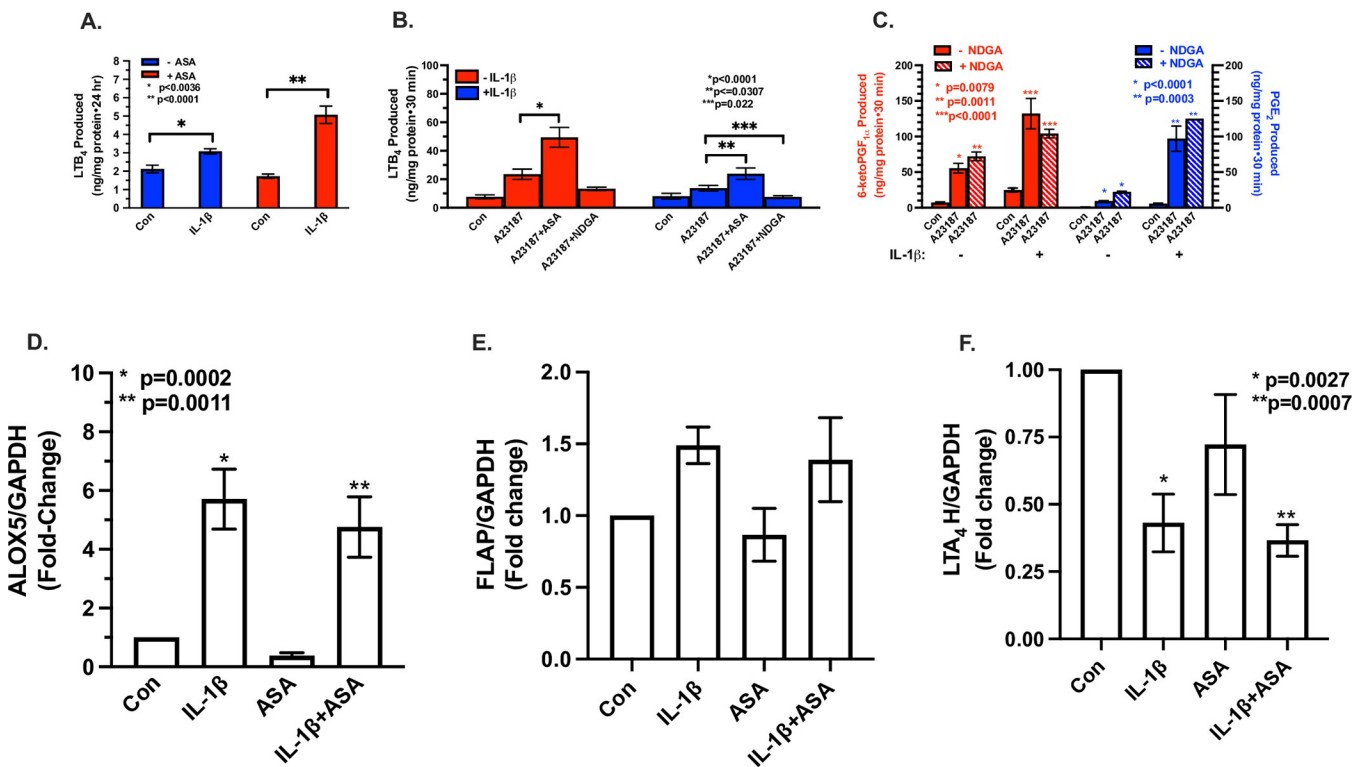

**Fig 3. IL−1β promotes LTB₄ biosynthesis in endothelial cells that is lipoxygenase-dependent and augmented by aspirin.** HUVEC were incubated in medium supplemented with 20 μM arachidonic acid (AA) and stimulated for 24 hr with IL−1β (2 ng/ml) or vehicle (PBS/0.1% BSA) in the presence or absence of ASA (0.1 mM). Media were analyzed for LTB₄ by ELISA. The data are the mean±SEM (n = 6–12), analyzed by two-tailed t-tests and the experiment was conducted twice. (A) Blue bars, no ASA treatment (p<0.0036), Red bars, ASA treatment (p<0.0001). (B-D) qRT-PCR analysis of (B) ALOX5, (C) FLAP and (D) LTA₄H. After a 24-hr incubation with IL−1β (2 ng/ml) in the presence or absence of ASA, total RNA was extracted and RT-PCR carried out using Taqman™ chemistry. Transcript expression was computed by the 2^{-ΔΔCt} method using GAPDH as a housekeeping reference gene and fold change relative to untreated controls was calculated. The data are the mean±SEM (n = 4–6 biological replicates) and the experiment was conducted twice.

control and IL−1β-stimulated cells (Fig 3B). Interestingly, aspirin treatment enhanced calcium ionophore-elicited LTB₄ production. As expected, PGI₂ and PGE₂ synthesis was resistant to NDGA in both unstimulated and IL−1β-stimulated endothelial cells (Fig 3C). Finally, we measured transcripts for the synthetic enzymes in the 5-LOX pathway responsible for LTB₄ synthesis and found that IL−1β-stimulated elicited a significant increase (p<0.0002) in ALOX5 mRNA expression, while the cytokine had no effect on FLAP and modestly inhibited LTA₄ synthase mRNA (Fig 3D–3F). We used HL-60 promyelocytic leukemic cells and THP-1 monocytic leukemic cells as positive controls for the expression of lipoxygenase encoding transcripts (Table 1). Aspirin co-incubation had no effect on IL−1β-stimulated endothelial cells for the expression of ALOX5, FLAP or LTA4H.

To extend the transcript expression data for ALOX5, FLAP and LTA4H, we conducted western blotting experiments to measure the cognate proteins. HUVEC were treated with IL−1β or vehicle in the presence or absence of aspirin and proteins were extracted, fractionated by SDS-PAGE and probed with LOX-5, FLAP or LTA4H antibodies. We detected an increase in 5-LOX protein expression (p = 0.048) (Fig 4A). Neither IL−1β alone nor in combination with aspirin caused a change in FLAP expression, although a modest decrease in LTA₄ hydrolase was detected in cells treated with IL−1β+ASA (Fig 4C, *p = 0.002). Most studies to date have indicated that 5-LOX is primarily expressed in cells of myeloid origin [28, 29]. As such, we used THP-1 monocytic cells differentiated into macrophage-like cells with phorbol myristate acetate (PMA, 200 ng/ml, 72 hr) and HL-60 promyeoloblastic cells differentiated in neutrophil-like

**Table 1. Ct values for gene transcripts.**

| Target | Gene Name | Amplicon Size (bp) | $C_t$ Value (HUVEC) | Ct Value (HL-60) | Ct Value (THP-1) |
|---|---|---|---|---|---|
| Cyclooxygenase-1 | PTGS-1, COX-1 | 60 | 27.5±4.04 | ND | ND |
| Cyclooxygenase-2 | PTGS-2, COX-2 | 75 | 29.6±2.38 | 25.56 | 24.73 |
| Prostacyclin Synthase | PGIS | 106 | 28.6±2.01 | ND | ND |
| Microsomal Prostaglandin E Synthase-1 | mPGES-1 | 68 | 35.2±1.90 | ND | |
| 5-Lipoxygenase | ALOX5 | 57 | 34.5±.912 | 22.36 | 13.7 |
| 12-Lipoxygenase | ALOX12 | 59 | 37.4±1.45 | 36.1 | 35.57 |
| 15-Lipoxygenase-1 | ALOX15 | 64 | 37.1±1.56 | 36.3 | ND |
| Leukotriene $A_4$ Hydrolase | LTA4H | 77 | 24.7±.600 | 21.58 | |
| 5-Lipoxyenase-activating protein | FLAP | 80 | 27.2±1.83 | 19.73 | 22.26 |
| Glyceraldehyde 3-phosphate dehydrogenase | GAPDH | 157 | 19.2±.157 | 16.58 | 12.95 |

Shown are the Ct values for the target genes analyzed (control, untreated). Values are mean±SEM for HUVEC from a minimum of 3 experiments. HL-60 and THP-1 cells were used as positive controls.

cells with 2% DMSO as positive controls for western blots prepared to confirm the presence of lipoxygenase pathway proteins in HUVEC (Supplemental Information, S1 Fig).

## Pro-inflammatory IL−1β elicits pro-resolving lipoxin synthesis in endothelial cells: Role of 5-lipoxygenase/FLAP pathway

We next examined whether endothelial cells could synthesize Lipoxin $A_4$ (LXA$_4$), the first discovered pro-resolving metabolite of arachidonic acid (AA) [30, 31]. HUVEC were cultured as described in complete media, and, after reaching confluence were incubated with IL−β (2 ng/ml) in the presence or absence of aspirin and LXA$_4$ was assayed by ELISA. Fig 5A shows that cytokine stimulation of endothelial cells caused nearly 3-fold increase in LXA$_4$ formation in both the presence ($p<0.005$) or absence of aspirin (**$p<0.05$). Aspirin-triggered lipoxin, (15-epi-LXA$_4$) was detected only in aspirin-treated cells (Fig 5B, *$p<0.005$). As expected, aspirin treatment of HUVEC caused a dose-dependent suppression of the COX products, PGI$_2$ (measured as 6ketoPGF$_{1\alpha}$) and PGE$_2$ (Fig 5C), which stimulated 15-epi-LXA$_4$ (R-enantiomer) but not LXA$_4$ (S-enantiomer) (Fig 5D).

To further investigate the mechanism of pro-inflammatory and pro-resolving lipid formation in endothelial cells, HUVEC stimulated with IL−1β or vehicle were pre-incubated with 20 μM arachidonic acid in the presence or absence of the selective 5-LOX inhibitor, Zileuton (0.2 μM), or the FLAP inhibitor, MK-886 (0.2 μM). Cells were then challenged with A23187 (50 μM) and media were analyzed for vasoactive PGE$_2$ and PGI$_2$ and SPMs LXA$_4$ and 15-epi-LXA$_4$. Fig 6A and 6B shows that aspirin inhibits PGI$_2$ and PGE$_2$ production. Interestingly, cells incubated with Zileuton (specific 5-LOX inhibitor) or MK-886 (FLAP inhibitor) led to a diminution in the synthesis of both eicosanoids, but only in cells that had been stimulated overnight with IL−1β (Fig 6A and 6B, right panels), presumably due to COX-2 induction. In contrast, both Zileuton and MK-886 profoundly diminished LXA$_4$ and 15-epi-LXA$_4$ synthesis, indicating a pivotal role for 5-LOX and FLAP.

## Aspirin suppresses pro-inflammatory eicosanoid synthesis in endothelial cells

Aspirin has been used for decades as an anti-thrombotic and anti-inflammatory agent, but direct actions on the endothelium have not been extensively studied. HUVEC were stimulated for 24 hr with IL−1β (0.2 or 2 ng/ml) in the presence or absence of aspirin ($10^{-4}$–$10^{-10}$ M). Media were collected and PGI$_2$ (measured as 6-ketoPGF$_{1\alpha}$) and PGE$_2$ analyzed by ELISA. For both IL−1β concentrations, aspirin caused a dose-dependent suppression of both eicosanoids

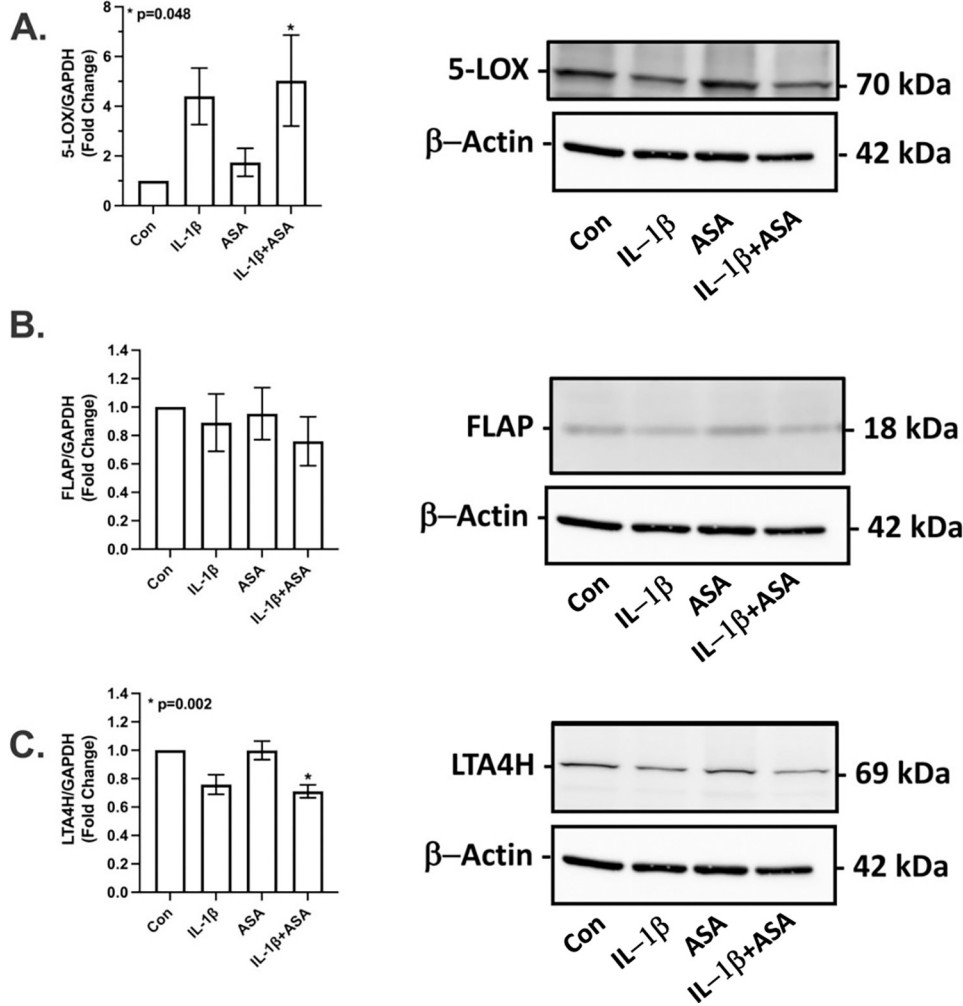

**Fig 4. Endothelial cells express enzymes in involved in leukotriene and lipoxin biosynthesis.** HUVEC were stimulated for 24 hr in the presence or absence of IL−1β (2 ng/ml) with or without ASA (0.1 mM). Proteins were fractionated by SDS-PAGE and western blots probed with antibodies directed against: (A) 5-LOX; (B) LTA4H; or (C) FLAP, developed by chemiluminescence and analyzed by densitometry. Blots were reprobed with β−Actin as a loading control. (D) Representative western blots. The data are reported as mean±SEM (n = 3 biological replicates) of target:β−Actin expression (A.U.) and analyzed by one-way ANOVA followed by Dunnett's multiple comparison test.

(Fig 7). The $IC_{50}$ for aspirin inhibition of $PGI_2$ was approximately an order of magnitude more potent than $PGE_2$ ($IC_{50}(PGI_2) \sim 4$ μM; $IC_{50}(PGE_2) \sim 0.2$ mM)

We next compared intact acetylsalicylic acid (aspirin) and non-acetylated salicylic acid (SA) in the eicosanoid suppression assay. HUVEC were challenged as above with IL−1β in the presence or absence of aspirin or SA ($10^{-4}$–$10^{-10}$ M). As shown in Fig 8, aspirin inhibited both $PGI_2$ and $PGE_2$ production in a dose-dependent manner, while SA was ineffective even at the highest concentration (0.1 mM).

## Discussion

### Principal findings

The endothelium interfaces with circulating immune cells and chemical mediators of inflammation in preeclampsia and other cardiovascular disorders [32, 33]. As such, endothelial cells

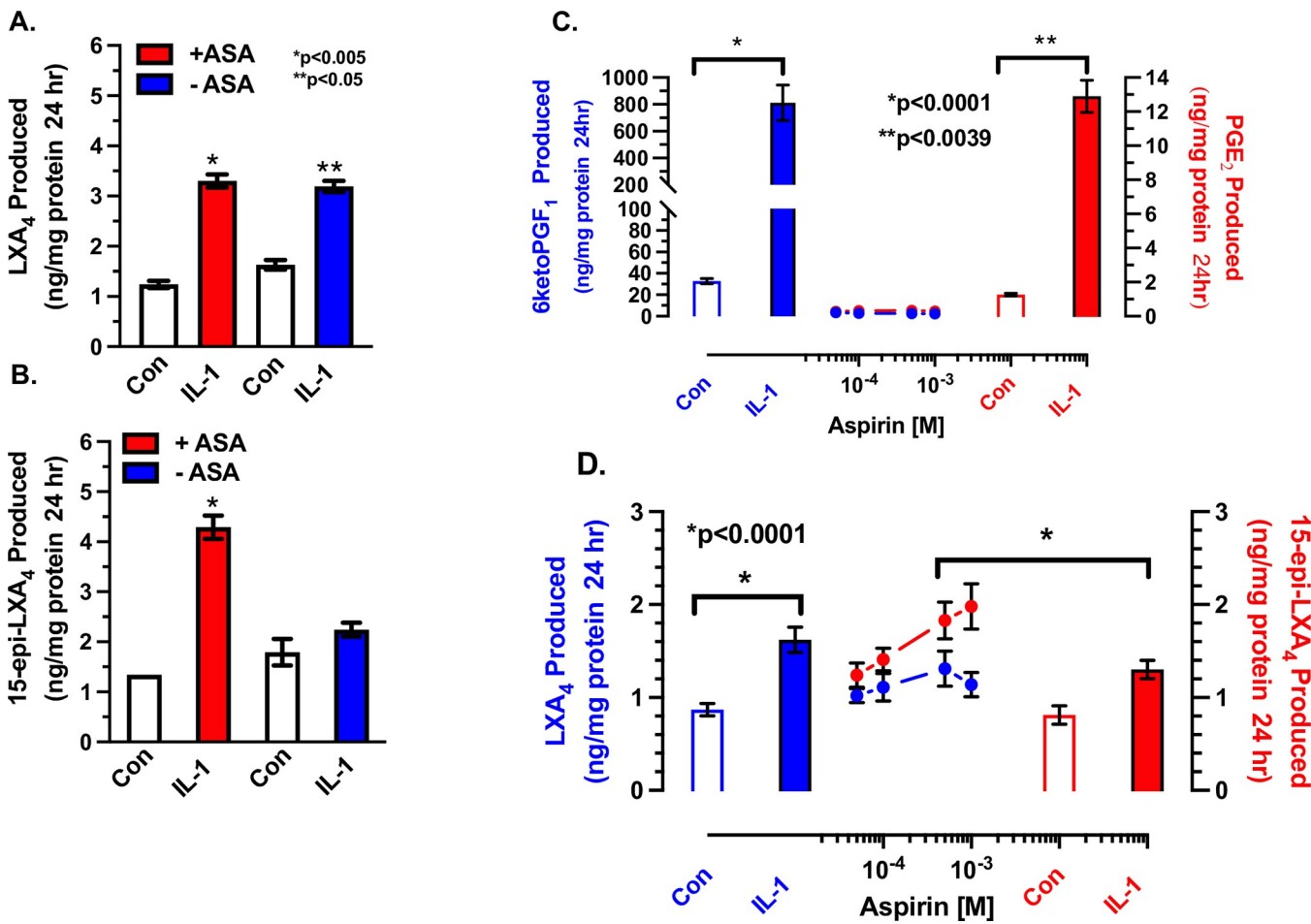

**Fig 5. IL−1β promotes LXA₄ synthesis and aspirin stimulates 15-epi-LXA₄ in endothelial cells in the absence of leukocytes.** HUVEC were stimulated for 24 hr with IL−1β (2 ng/ml) or vehicle (PBS/0.1% BSA) in the presence or absence of ASA (0.1 mM). Media collected were analyzed for LXA₄ (**A**) and 15-epi-LXA₄ (**B**) by ELISA. The data are expressed as mean±SEM (ng/mg protein, n = 6–9 samples/condition) and analyzed by paired Student's t-test (*p<0.005; **p<0.05). C and D show the aspirin dose-response effects on IL−1β-stimulated PGI₂, PGE₂ (**C**) and LXA₄, 15-epi-LXA₄ (**D**). Data are representative of two experiments.

targeted by proinflammatory signals produce bioactive lipids derived from arachidonic acid (AA, 20:4, ω-6) (prostaglandins, prostacyclin, thromboxane, leukotrienes, cysteinyl leukotrienes and cytochrome P₄₅₀ metabolites of AA) that act locally and systemically to orchestrate a coordinated immune response to external threats to homeostasis [34–36]. In the current work, we demonstrated a dose-dependent increase in PGI₂ and PGE₂ formation in response to both IL−1β and TNFα, two pivotal cytokines that are involved in nearly all forms of acute inflammation [37–40]. This effect was driven by a robust induction of COX-2, the rate-limiting enzyme, and constitutive expression of PGIS and mPGES-1, the terminal enzymes that produce PGI₂ and PGE₂, respectively.

We also found that endothelial cells produced the lipoxygenase product LTB₄ following cytokine stimulation in the absence of leukocytes, a novel finding given the transcellular hypothesis of leukotriene synthesis during inflammation [25, 41]. Leukotrienes produced by endothelial cells have previously been thought to be produced only in cooperation with other neutrophils or monocyte/macrophages, since they are thought to lack 5-LOX [25, 42]. However, our data suggest that umbilical cord endothelial cells express low, albeit detectable levels of ALOX5 mRNA and the cognate 5-LOX protein (see Figs 3 and 4). These results are

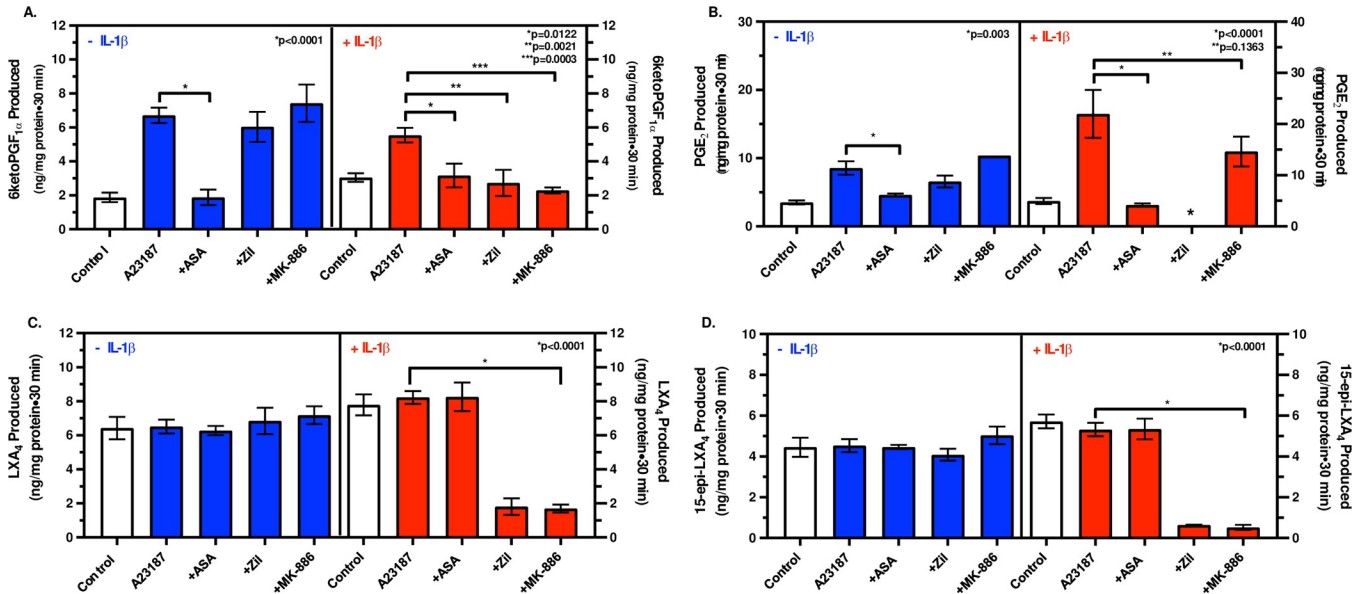

**Fig 6. 5-LOX/FLAP pathway inhibitors attenuate pro-inflammatory (A, B, top panel) and pro-resolving (C, D, lower panel) lipids.** HUVEC were stimulated for 24 hr with IL−1β (2 ng/ml, blue bars) or vehicle (PBS/0.1% BSA, red bars). Cells were then preincubated for 60 min with 20 μM arachidonic acid in the presence or absence of inhibitors (aspirin, 0.2 mM, Zileuton, 0.2 μM or MK-886, 0.2 μM) followed by a 30-min challenge with calcium ionophore (A23187, 50 μM). Lipid production was measured by ELISA. The data are mean±SEM (n = 6–12 samples/treatment) and the data were analyzed by one-way ANOVA followed by Tukey's test for differences. The data are representative of two experiments. * Denotes that $PGE_2$ level was below Lower Limit of Quantification (LLOQ).

supported by a similar report by Chatterjee et al. [43]. Interestingly, we noted an increase in IL−1β-induced ALOX5 and FLAP mRNAs, but a decrease in LTA4 in cytokine-treated cells (Fig 3D–3F). These results indicate that elevated $LTB_4$ in HUVEC resulted from increased availability of $LTA_4$ and not induction of the terminal leukotriene-synthesizing enzyme. Moreover, our data in Fig 3B showing suppression of $LTB_4$ by the lipoxygenase inhibitor, NDGA, supports the observation of endothelial cell formation of LT in the absence of leukocytes. The cause of this apparent discrepancy is not currently known but is the subject of investigation.

Our use of highly sensitive ELISA rather than mass spectrometry-based measurement which is accurate but less sensitive may be a partial explanation. In addition, we detected appreciable amounts of $LTB_4$ only in cells exposed to cytokine see Fig 3A). Furthermore, the finding of elevated $LTB_4$ in cells pretreated with aspirin suggests substrate shunting toward the lipoxygenase pathway [44]. These data may have implications for a direct role of endothelial cells in producing leukocyte chemoattractant lipids during inflammation [42]. Considering the current data, we have proposed a reappraisal of the model of lipoxygenase products ($LTB_4$ and Lipoxins $A_4$ (both S- and R- enantiomers)) made by endothelial cells (Fig 9). In contrast to the conventional view that endothelial cells purportedly lacking 5-LOX, cannot produce lipoxygenase products independent of leukocytes or platelets, our *in vitro* experiments indicate that, indeed, human endothelial cells stimulated by cytokines are capable of leukotriene and lipoxin biosynthesis without the aid of accessory cells. Furthermore, aspirin exerts direct effects on endothelial cells by driving the production of 15-epi-$LXA_4$, a highly potent pro-resolving mediator.

## Aspirin exhibits pleiotropic actions in endothelial cells

Low-dose aspirin is used extensively as a preventive agent for adult cardiovascular and thrombotic diseases [45, 46]. Moreover, currently aspirin is the only recommended drug

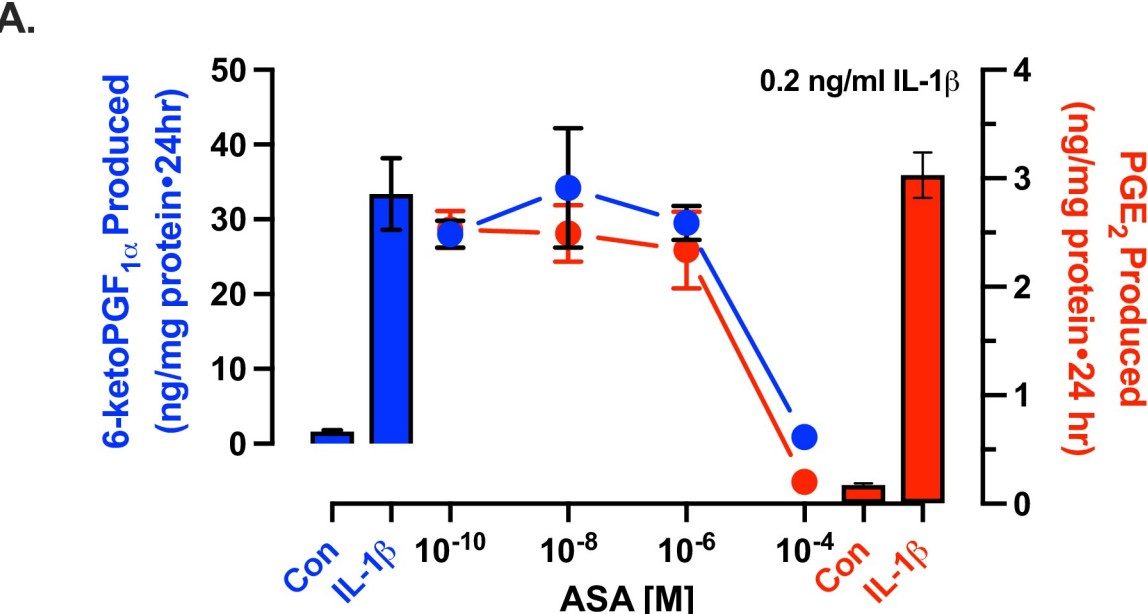

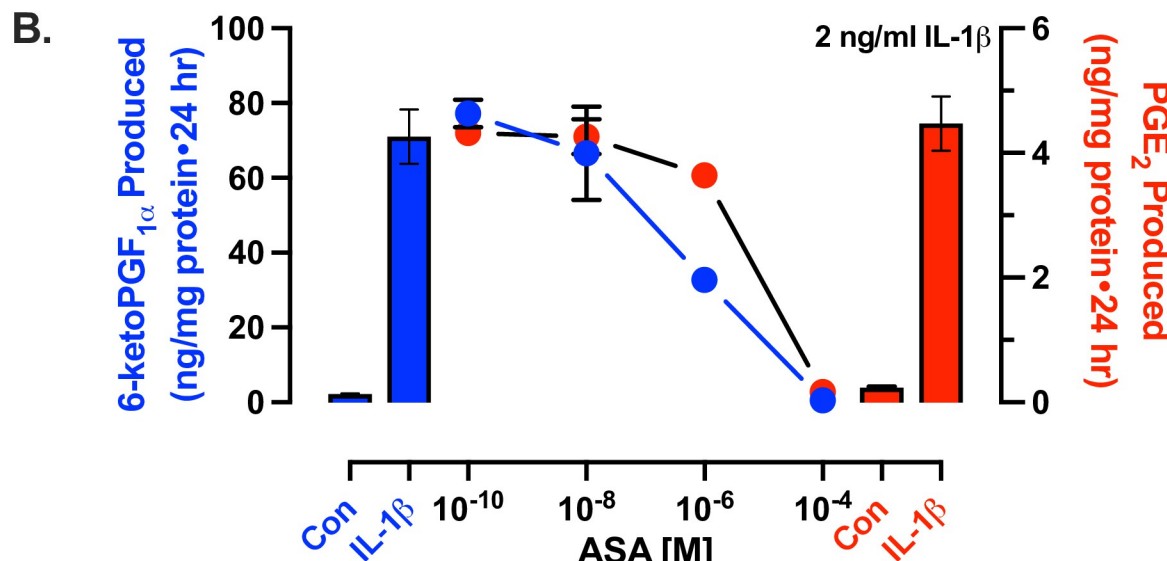

**Fig 7. Aspirin inhibits synthesis of pro-inflammatory, vasoactive eicosanoids in endothelial cells.** HUVEC were stimulated with IL−1β (A, 0.2 ng/ml or B, 2 ng/ml) M199+0.5% charcoal-stripped FCS for 24 hr in the presence of ASA (10−4–10−10 M) or vehicle (0.1% EtOH) and media were analyzed for PGI$_2$ (6-ketoPGF$_{1\alpha}$ and PGE$_2$) by ELISA. The data are expressed as mean±SEM (ng/mg protein, n = 6–12 samples/concentration). Experiment was replicated three times.

administered early in pregnancy for women at risk for developing preeclampsia [47, 48]. The mechanism action of aspirin is the subject of intensive investigation and involve the regulation of several signaling pathways in the inflammatory cascade [49–51]. As a classical NSAID, aspirin directly inhibited pro-inflammatory eicosanoid (PGI$_2$ and PGE$_2$) formation in endothelial cells. In contrast, when cells were induced with IL−1β to express COX-2, aspirin increased the

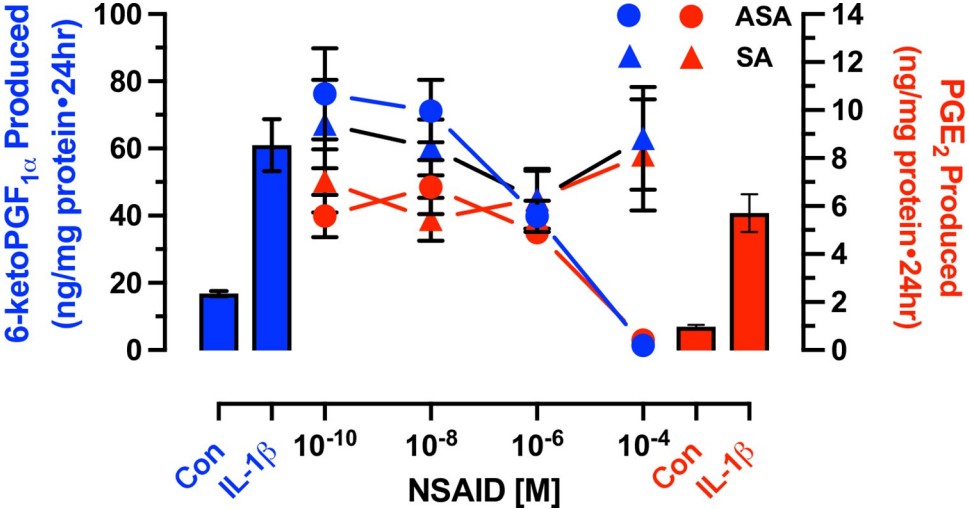

**Fig 8. Sodium salicylate does not inhibit pro-inflammatory eicosanoids in endothelial cells.** HUVEC were stimulated with IL−1β (2 ng/ml) M199+0.5% charcoal-stripped FCS for 24 hr in the presence of $10^{-4}$–$10^{-10}$ M ASA (circles), SA (triangles) or vehicle (0.1% EtOH) and media were analyzed for $PGI_2$ (6-ketoPGF$_{1\alpha}$ and $PGE_2$) by ELISA. The data are expressed as mean±SEM (ng/mg protein, n = 6–12 samples/concentration). Experiment was replicated two times.

formation of pro-resolving LXA$_4$ and 15-epi-LXA$_4$. These results, taken together, support the beneficial anti-inflammatory, pro-resolving function of low-dose aspirin, and suggest endothelial cells are a direct producer of both pro-inflammatory and pro-resolving mediators. However, when cells were preincubated with aspirin prior to short-term challenge with calcium ionophore (A23187), there was a significant increase in LTB$_4$ production, suggesting that aspirin may also exacerbate neutrophil chemotaxis and thereby enhance inflammation by shunting arachidonic acid toward the lipoxygenase pathway in endothelial cells. In addition, the ability of even low doses of aspirin to inhibit PGI$_2$ synthesis in endothelial cells indicate a potential undesirable effect on vasodilator production. Thus, collectively our study may prompt a reevaluation of the exclusively beneficial effects of aspirin in vascular biology [20, 21, 52, 53].

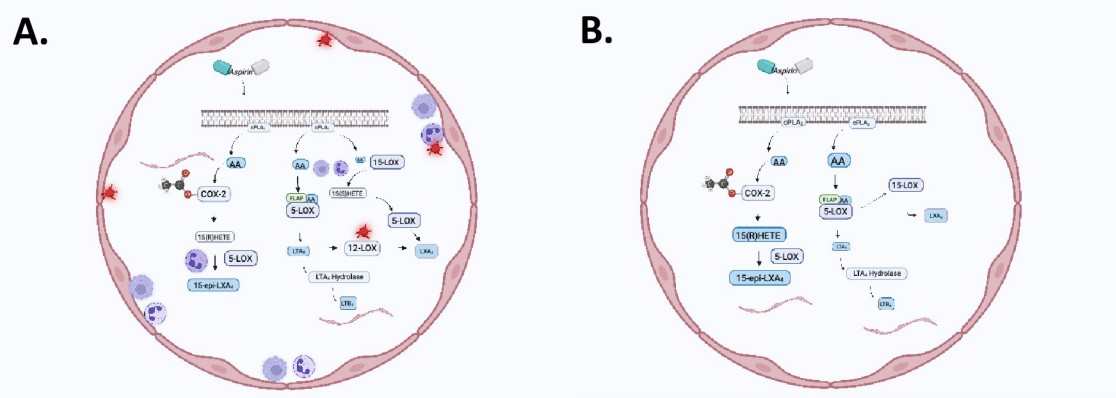

**Fig 9. Proposed model of single cell synthesis of LTB$_4$ and LXA$_4$ in endothelial cells.** The conventional model suggests that endothelial cells and leukocytes (infiltrating neutrophils or monocyte/macrophages or platelets) cooperate in transcellular synthesis of chemoattractant LTB$_4$ and pro-resolving LXA$_4$ during acute inflammation **(A)**. Our updated model suggests that endothelial cells express modest levels of the enzyme machinery necessary for leukocyte/platelet-independent lipid synthesis **(B)**.

## Study limitations

Like all biomedical research, our investigations suffer from a few limitations. First, all *in vitro* studies must be interpreted with caution when extrapolating to the clinical setting. Second, we used human umbilical endothelial cells (HUVEC) as a convenient, readily isolated model system for our experiments. While HUVEC are frequently used in studies of endothelial cell biology, it is now clear that heterogeneity exists in endothelial cells depending on location within the vascular tree (organ-specific differences), caliber (i.e., derived from large and small vessels or microvasculature) and physiological state and pathological conditions [54–57]. Moreover, arterial and venous endothelial cells differ phenotypically due to distinct biomechanical micro-environments [58–61]. The use of HUVEC, cells of fetal origin, may not represent the exact phenotype of maternal endothelial cells that are the target of dysfunction in preeclampsia. However, obtaining endothelial cells from maternal vessels is ethically challenging, and HUVEC provide a reasonable model for fundamental studies.

In the present studies, we used highly sensitive commercial ELISAs to measure lipids, while most reports employed LC-MS/MS based methods. LC-MS/MS is very accurate in identifying the molecular identification of biological compounds but suffers from lower sensitivity than immunological methods. Thus, there is a trade-off between sensitivity on the one hand (ELISA) and exquisite accuracy but lower sensitivity (LC-MS/MS) on the other hand [12, 62–65]. The assays used in the present work are highly validated, and we were able to discriminate the S- and R- enantiomers of $LXA_4$ (see Fig 5). The ideal investigation of bioactive lipids combines the assets of both ELISA for high sensitivity and LC-MS/MS for highly accurate and potentially unbiased identification of both known and unknown moieties.

## Conclusions and future directions

Our investigations suggest the intriguing possibility that endothelial cells can synthesize vaso-active eicosanoids and chemoattractant ($PGI_2$, $PGE_2$ and $LTB_4$) lipids and pro-resolving $LXA_4$ and aspirin-triggered $LXA_4$ independent of leukocytes or platelets.

These findings offer a new twist pro-inflammatory/pro-resolving cascade during acute inflammation and its eventual resolution (Fig 9). Previous models suggest that endothelial cells are reliant on local leukocytes or platelets for lipid production, especially SPMs and LTs via transcellular biosynthesis, largely because they lack all the necessary enzymatic machinery for independent production [23, 66]. We are examining whether women with preeclampsia show altered expression of the enzyme machinery necessary to produce pro-resolving $LXA_4$ and its stereoisomer, 15-epi-$LXA_4$, in response to low-dose aspirin as a potential mechanism for their clinical manifestations of endothelial dysfunction during pregnancy [67].

The overall implications of our investigations suggest that endothelial cells are far more versatile in their ability to synthesize both pro-inflammatory and pro-resolving lipid mediators independent of accessary cells, e.g., neutrophils, macrophages or platelets, than previously appreciated. It will be important in future studies to define more thoroughly the enzymology of bioactive lipid biosynthetic pathways in isolated endothelial cells from a variety of sources and conditions (e.g., large vessel endothelia, microvascular endothelia, venous versus arterial endothelia, healthy versus diseased) given the increasing interest in developing therapeutic agents to control lipid formation in the context of inflammation in cardiovascular diseases including preeclampsia.

## Supporting information

**S1 Fig. Western blot analysis of lipid synthesizing enzymes in the lipoxygenase pathway.** (5-lipoxygenase, 5-LOX; 5-lipoxygenase-activating protein, FLAP; leukotriene $A_4$ hydrolase,

LTA4H; 12-lipoxygenase, 12-LOX; 15-lipoxygenase-1, 15-LOX-1; β–Actin). Lane 1, PMA-differentiated THP-1 cells; lane 2, DMSO-differentiated HL-60 cells; lane 3, unstimulated HUVEC. Cell extracts (30 μg/lane) were fractionated on 4–20% SDS-PAGE, transferred to nitrocellulose and probed with antibodies (see S1 **Table**).
(PPTX)

**S1 Table. Antibodies.** Polyclonal or monoclonal antibodies were used at the indicated dilutions. Where possible, positive controls (THP-1, HL-60 cell lysates for 5-LOX, 15-LOX, LTA4H and FLAP proteins) or recombinant proteins for COX-1 and COX-2. In addition, negative controls also consisted of pre-absorption of primary antibodies with blocking peptides where available.
(DOCX)

**S2 Table. PCR primer sets.** All qRT-PCR experiments were conducted using Taqman™ chemistry and data were analyzed using the $2^{-\Delta\Delta Ct}$ method.
(DOCX)

**S3 Table. M199 media supplemented with 0.5% charcoal-stripped serum (0.5% FCS) and 5% Endothelial Cell Growth Supplement (5% ECGS) not exposed to cells was analyzed for lipid analytes listed in the table.** All analytes were below the lower limit of detection. Duplicate samples were analyzed in two separate experiments.
(DOCX)

## Author Contributions

**Conceptualization:** Kara M. Rood, Ivana M. DeVengencie, Maged M. Costantine, Douglas A. Kniss.

**Data curation:** Niharika Patel, Ivana M. DeVengencie, John P. Quinn, Douglas A. Kniss.

**Formal analysis:** Douglas A. Kniss.

**Funding acquisition:** Kara M. Rood, Maged M. Costantine.

**Investigation:** Kara M. Rood, Niharika Patel, John P. Quinn, Douglas A. Kniss.

**Methodology:** Niharika Patel, Ivana M. DeVengencie, Kymberly M. Gowdy, Douglas A. Kniss.

**Project administration:** Maged M. Costantine.

**Resources:** Kara M. Rood, Maged M. Costantine.

**Supervision:** Douglas A. Kniss.

**Validation:** Kymberly M. Gowdy.

**Visualization:** Kymberly M. Gowdy.

**Writing – original draft:** Douglas A. Kniss.

**Writing – review & editing:** Kara M. Rood, Ivana M. DeVengencie, John P. Quinn, Kymberly M. Gowdy, Maged M. Costantine.

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
