## [Decision Letter · Decision Letter 0]

29 Aug 2022

PONE-D-22-15963Aspirin modulates expression of pro-inflammatory and pro-resolving mediators in endothelial cellsPLOS ONE

Dear Dr. Kniss,

Thank you for submitting your manuscript to PLOS ONE. After careful consideration, we feel that it has merit but does not fully meet PLOS ONE’s publication criteria as it currently stands. Therefore, we invite you to submit a revised version of the manuscript that addresses the points raised during the review process, in particular the point highlighted below:

Reviewers raised concerns about the specificity of ELISA assays to unequivocally  determine the production of specific lipid mediators. Further information regarding cross reactivity and sensibility of the assays should be included in the manuscript. The analyses should be complemented by direct determination by LC-MS/MS and/or by use of selective genetic/pharmacological tools.

15-LO expression should be directed determined in HUVEC and preferably in some fashion that allow directly association of enzyme expression and the cell s containing endothelial markers  such as immunofluorescence.

We look forward to receiving your revised manuscript.

Kind regards,

Bruno Lourenco Diaz, Ph.D.

Academic Editor

PLOS ONE

https://journals.plos.org/plosone/s/file?id=ba62/PLOSOne_formatting_sample_title_authors_affiliations.pdf".

“This work was supported in part by an intramural research grant from the Center for Clinical and Translational Science (NIH NCATS, No. UL1TR002733) to KMR. “

“Award, KMR

Grant No. UL1TR002733

Center for Clinical and Translational Sciences, Ohio State University/NIH NCATS

ccts.osu.edu; nih.gov

Sponsor played no role in the study design, data collection and analysis, decision to publish, or preparation of the manuscript”

“No authors have competing interests”

4. PLOS requires an ORCID iD for the corresponding author in Editorial Manager on papers submitted after December 6th, 2016. Please ensure that you have an ORCID iD and that it is validated in Editorial Manager. To do this, go to ‘Update my Information’ (in the upper left-hand corner of the main menu), and click on the Fetch/Validate link next to the ORCID field. This will take you to the ORCID site and allow you to create a new iD or authenticate a pre-existing iD in Editorial Manager. Please see the following video for instructions on linking an ORCID iD to your Editorial Manager account: https://www.youtube.com/watch?v=_xcclfuvtxQ.

7. Please include your tables as part of your main manuscript and remove the individual files. Please note that supplementary tables (should remain/ be uploaded) as separate ""supporting information"" files.

Reviewers' comments:

Reviewer's Responses to Questions

**Comments to the Author**

1. Is the manuscript technically sound, and do the data support the conclusions?

Reviewer #1: Partly

Reviewer #2: Yes

Reviewer #3: Partly

2. Has the statistical analysis been performed appropriately and rigorously? 

Reviewer #1: Yes

Reviewer #2: Yes

Reviewer #3: Yes

3. Have the authors made all data underlying the findings in their manuscript fully available?

Reviewer #1: No

Reviewer #2: Yes

Reviewer #3: Yes

4. Is the manuscript presented in an intelligible fashion and written in standard English?

Reviewer #1: Yes

Reviewer #2: Yes

Reviewer #3: Yes

5. Review Comments to the Author

Reviewer #1: This manuscript describes that HUVECs are capable of producing not only prostaglandins on their own but that they also have the enzymes necessary for the production of LTB4 and LXA4. In addition the authors describe experiments showing that aspirin treatment of HUVECs shifts the production of prostaglandins towards LTB4 and LXA4 / AT-LXA4 in a COX-2 dependent pathway.

The manuscript is well written and corroborates the current knowledge of the biosynthesis of these lipid mediators. Novel is, however, that all of these pathways co-occur in just one cell type. It is therefore essential to show that the measurements of the lipid mediators is specific and that not other lipid mediators are co-measured with the assays.

The authors may consider the following criticism and open questions to their manuscript.

1. Are the authors sure that the immunoassays applied to measure lipid mediators in complex mixtures of cell extract are specific for the indicated lipid mediator. They should verify their results by LC-MS analyses. Current LC-MS assays are more sensitive than the immunoassays (e.g. https://doi.org/10.1515/cclm-2021-0644).

2. Each immunoassay has a measuring range and a LLOQ. Are all data points included in the manuscript for all lipid mediators within the measuring range of the respective immunoassay?

3. While expression analyses have been done for most enzymes necessary for the lipid mediator production investigated in endothelial cells, the expression of ALOX15, necessary for LXA4 production, is not shown in the manuscript.

4. What levels of lipid mediators are present in the medium containing FCS used to incubate the HUVECs? How can the authors exclude the incorporation of lipid mediators or their precursors into HUVECs when using FCS in the cell culture medium?

5. mPGES-1 protein expression is not visible on the western blot in Fig. 1H.

6. The results for the qPCR analysis are shown as fold increase only. However, it would be interesting to see how low the baseline levels for the different mRNAs are. A doubling of very low levels may not indicate a biologically relevant increase.

7. There are no figure legends included in the manuscript, which makes the interpretation of the figures hard.

8. It is not clear whether COX-1 and COX-2 are depicted in Figure 2A).

9. The western blot depicted in figures 4 A) and B) do not support the results depicted in the respective graphs.

10. Second last line on page 9 should be written Fig. 1C and not Fig. 1B.

11. Figure 8 is not readable due to a too low resolution of the TIF files.

Reviewer #2: I read with interest this original research article. The concept of this manuscript is valid and the presentation adequate. A few grammar should be addressed.

In the method section in the cytokines used please clarified why IL-6 is referred 2 times.

The limitations section provide a balance description of the weak points of this work.

Please discuss the role of aspirin in low and high dose regarding anti-inflammatory and anti0thrombotic actions

In the results section pleased avoid to discuss your finding or to cite bibliography or to described the design of the study.

The conclusion section should be shorten

Reviewer #3: The article by Rood et al. entitled “Aspirin modulates expression of pro-inflammatory and pro-resolving mediators in endothelial cells” presents results demonstrating that human umbilical vein endothelial cells (HUVEC) biosynthesize both pro-inflammatory and pro-resolving mediators in absence of participation from leukocytes. The manuscript is well-written overall and some of the results are novel, but there are additional experiments that should be carried out to solidify the conclusions made. I have a few comments and suggestions that the authors may want to consider to strengthen the manuscript.

1. As the authors point out, 5-LOX is typically not highly expressed in endothelial cells, although some evidence suggests that it can be induced under certain conditions (e.g., cytomegalovirus infection, hypoxia, etc). As such, it would be helpful if additional experiments could be performed to further solidify the conclusion that 5-LOX is expressed in HUVEC and that it contributes to the biosynthesis of leukotrienes and lipoxins. For example, it would be helpful to include a positive control for 5-LOX expression (e.g., human neutrophils that express high levels of 5-LOX) and to use siRNA for Alox5 as a negative control when assessing its expression by Western blot to validate specificity of the antibody.

2. The results show that native (15S) lipoxin A4 is produced in the HUVEC, implying expression of 15-LOX in this particular lot of cells (pooled donors I presume). Have the authors evaluated expression of 15-LOX type 1 and 2 in these cells? Is its production inhibited by selective 15-LOX inhibition or siRNA?

3. The inhibitor used, NDGA can be non-specific, inhibit several lipoxygenases, and potentially have anti-oxidant/free radical scavenging effects. Formation of non-enzymatic lipid oxidation products could occur in these cells under inflammatory conditions and structurally similar non-enzymatic oxidized lipids could potentially be recognized by the ELISAs used to detect leukotrienes for example (see point 4 below). Thus, a more specific approach like siRNA (as described in Point 1) or a distinct inhibitor should additionally be used to definitively document the enzymatic origin of leukotrienes and lipoxins in these cells.

4. (Related to Point 3): ELISAs for lipid mediators are known to have cross reactivity. For instance, the kits for LXA4 and 15R-LXA4 show cross reactivity for resolvin D1 and 17R-resolvin D1, respectively, while the kit for LTB4 shows cross reactivity for both 5S-HETE and 5R-HETE (which could be formed during oxidative stress non-enzymatically as a racemic mixture). As such, since a more specific approach such as LC-MS/MS is not being used, the authors should at least more thoroughly establish the enzymatic origin of these mediators using additional distinct and selective inhibitors of 5-LOX in cells co-treated with ASA for 15R-LXA4, and inhibitors of FLAP, LTA4H, and a distinct inhibitor of 5-LOX (other than NDGA) for LTB4, as noted above.

5. There are no figure legends are present in the manuscript and they should be included in a revision.

6. PLOS authors have the option to publish the peer review history of their article (what does this mean?). If published, this will include your full peer review and any attached files.

Reviewer #1: No

Reviewer #2: No

Reviewer #3: No

---

## [Author Response · Author response to Decision Letter 0]

7 Dec 2022

Responses to Reviewers’ Comments:

General comments:

1. Concern about the specificity of ELISA assays to unequivocally determine the production of specific lipid mediators. Response: We recognize that the gold standard for the measurement of lipids in biological matrices is LC-MS/MS, typically using triple quadrupole mass spectrometry. In the current work, we used highly specific and thoroughly validated ELISAs from Cayman Chemical Co. to measure (6keto PGF1a, the stable metabolite of PGI2, PGE2, LTB4, LXA4 and 15-epi-LXA4 (Aspirin-triggered Lipoxin A4). Each of these solid-phase immunoassays has been used by many authors in high-profile publications. I have had in-depth discussions with the scientific staff at Cayman Chemical regarding any nuances with these assays. Each assay has been extensively tested for cross-reactivity (this information is included in the Supplementation Information included in the revised manuscript. We did not resort to LC-MS/MS in this particular manuscript due to financial limitations. Moreover, the use of 96-well ELISA formats permits the execution of large numbers of biological replicates that can be subjected to statistical testing. It would have been cost-prohibitive to have done the same number of replicates using LC-MS/MS. (My lab is working carefully with the Lipidomics Core facility at Wayne State University in Detroit, MI and with Cayman Chemical Co, Ann Arbor, MI to confirm the authenticity, accuracy, precision, specificity and sensitivity of the ELISA in comparison to conventional LC-MS/MS.). Dr. Gowdy has recently published a paper that demonstrated strong agreement between ELISA and LC-MS/MS values for PGE2 in lung tissue specimens (Yeager et al., Toxicol Sci, 2021; 183: 170, PMID: 34175951). Finally, we have conducted new experiments using pharmacological inhibitors of key enzymes and proteins in the lipid synthesizing pathways (e.g., 5-lipoxygenase inhibitor, Zileuton; FLAP inhibitor, MK-886) to confirm the results and conclusions in the original manuscript submission. 

2. 15-LO expression should be directed determined in HUVEC and preferably in some fashion that allow directly association of enzyme expression and the cell s containing endothelial markers such as immunofluorescence. Response: The culture system used in this work is HUVEC that are 100% endothelial in nature. There is no evidence of contamination from any other cell type. Thus, although it cannot be concluded that equal expression of all lipid synthesizing enzymes is found in each and every cell in the cultures, nonetheless, it is valid to conclude that whatever enzyme expression is detected (by qRT-PCR and western blotting) is the located in endothelial cells. 

Specific responses to reviewers’ comments:

1. Are the authors sure that the immunoassays applied to measure lipid mediators in complex mixtures of cell extract are specific for the indicated lipid mediator. They should verify their results with LC-MS analyses. Current LC-MS assays are more sensitive than the immunoassays. Response: 1. We are familiar with the attached paper outlining LC-MS/MS methods. We agree that LC-MS/MS continues to be the gold standard for lipidomics. In the current work we made the calculated decision to use ELISA methods in order to conduct many experiments using biological replicates. This would have been cost-prohibitive if LC-MS/MS were the selected method, since each sample costs approximately $100 to prepare and analyze. The ELISA assays used in this work were from Cayman Chemical Co. and have been extensively validated by the company and in-house for specificity and sensitivity. More detail on our in-house validation is presented in the Supplemental Information section of this revised manuscript. I have had extensive discussions with the scientific staff at Cayman Chemical and with Dr. Krishna Rao Maddipati, Director of the Lipidomics Core at Wayne State University regarding the pros and cons of ELISA versus LC-MS/MS. While LC-MS/MS is highly specific in identifying the precise structure of lipids, it is less sensitive than ELISA on a molar basis. Moreover, it is typically quite sensitive the matrix used. In the current work, we did not use cell extracts as implied by the reviewer’s question, rather we collected culture media. The medium was supplemented only with 0.5% charcoal-stripped FBS and endothelial cell growth supplement (from bovine brain extract). We tested the serum and supplement for the lipids in question and they were at or below the limit of detection. Thus, lipids measured in the media represent those produced by HUVEC and not contaminants from the supplements. Finally, the ELISAs for each lipid analyzed showed minimal cross-reactivity with related compounds at the physiological concentrations found in the media. We will continue to compare the ELISA assays with LC-MS/MS, but, in this work, we are confident in the methods of analysis. 

2. Each immunoassay has a measuring range and a LLOQ. Are all data points included in the manuscript for all lipid mediators within the measuring range of the respective immunoassay? Response: Indeed, all reported values for lipids in the manuscript exceed the LLOQ. We were very careful about this assertion. 

3. While expression analyses have been done for most enzymes necessary for lipid mediator production investigated in endothelial cells, the expression of ALOX15, necessary for LXA4 production is not shown in the manuscript. Response: In the revised manuscript, we have included qRT-PCR and western blots showing expression of 15-LOX-1, including the use of THP-1 and HL-60 cell extracts as positive controls. 

4. What levels of lipid mediators are present in the medium containing FCS used to incubate the HUVECs? Response: For all experiments, 0.5% charcoal-stripped FBS was used. We measured the levels of each lipid mediators in the 0.5% FCS and they are at or below the LLOQ for each ELISA. 

5. mPGES-1 protein expression is not visible on the western blot in Fig. 1H. Response: In the revised manuscript, we have included a better western blot image to illustrate the expression of mPGES-1, the enzyme that synthesizes PGE2. 

6. The results of the qPCR analysis are shown as fold increase only. However, it would be interesting to see how low the baseline levels for the different mRNAs are. Response: For the transcript expression experiments, we used the well-accepted 2-��Ct method of Schmittgen & Livak (Nature Protocols, 2008; 3: 1101). This method takes into account the change in target gene relative to the expression of a reference gene based upon the relative Ct values of each transcript. However, Table 1 in the original submission also includes the absolute Ct values for ALOX5, LTA4H and FLAP. In the revised manuscript, we also include the Ct values for the other transcripts were measured. 

7. There are no figure legends included in the manuscript, which makes the interpretation of the figures hard. Response: We sincerely apologize for this oversight. Somehow, the figure legend was not uploaded in the original submission. We have included the figure legend in the revised manuscript. 

8. It is not clear whether COX-1 and COX-2 are depicted in Figure 2A. Response: Apologies for the lack of clarity. The y-axis for COX-2 is on the left (with blue type) and the y-axis for COX-1 is on the right (with red type) of Figure 2A. We used color coding of COX-2 and COX-1 on the same graph to illustrate the inducibility of COX-2 and the lower level, constituent expression of COX-1. 

9. The western blot depicted in figures 4 A) and B) do not support the results depicted in the respective graphs. Response: We have prepared new western blots and densitometric analysis and have inserted these new data into Figure 4. 

10. Second last line on page 9 should be written Fig. 1C and not Fig. 1B. Response: We thank the reviewers for pointing out this error. We have made the correction in the text. 

11. Figure 8 is not readable due to a too low resolution of the TIF files. Response: These imaging glitches have been corrected. 

Reviewer #2: … a few grammar should be checked. Response: Done.

1. Please discuss the role of aspirin in low and high dose regarding anti-inflammatory and anti-thrombotic actions. We made a few comments about the use of low-dose aspirin for preeclampsia versus the of aspirin as anti-platelet therapy. 

2. In the results section pleased avoid to discuss your finding or to cite bibliography or to described the design of the study. Response: The only instance in which we cited previous publications in the Results section (see under IL-1b stimulates LTB4 synthesis in endothelial cells independent of leukocytes: Modulation by aspirin) is relevant and supports the conduct of the experiment and has been retained in the revised manuscript. 

3. The conclusion should be shorten. Response: Per the reviewer’s suggestion, we have shortened the Conclusions and Future Directions section. 

Reviewer #3: 

1. As the authors point out, 5-LOX is typically not highly expressed in endothelial cells, although some evidence suggests that it can be induced under certain conditions (e.g., cytomegalovirus infection, hypoxia, etc). Response: In the revised manuscript, we have included positive control THP-1 (monocyte/macrophage) and HL-60 (neutrophil) cell lysates for lipoxygenase western blots. 

2. The results show that native (15S) lipoxin A4 is produced in the HUVEC, impression of 15-LOX in this particular lot of cells (pooled donors I presume). Have the authors evaluated expression of 15-LOX type 1 or 2 in these cells? Response: In the revised manuscript, we include western blots of 15-LOX-1 and 15-LOX-2 expression in HUVEC. It is worthy noting that the HUVEC cultures were from pooled donors and several separate batches of cells were used in the experiments. 

3. The inhibitor used, NDGA can be non-specific, inhibit several lipoxygenases, and potentially anti-oxidant/free radical scavenging effects. Formation of non-enzymatic lipid oxidation products could occur in these cells under inflammatory conditions and structurally similar non-leukotrienes for example (see point 4 below). Thus, a more specific approach like siRNA (as described in Point 1) or a distinct inhibitor should additionally be used to definitively document the enzymatic origin of leukotrienes and lipoxins in these cells. Response: In the revised manuscript, we have included Zileuton, a specific 5-LOX inhibitor and MK-886, a FLAP inhibitor, in addition to NDGA, to document the specific LOX pathways involved in the production of leukotrienes and lipoxins in endothelial cells. 

4. (Related to Point 3): ELISAs for lipid mediators are known to have cross reactivity. For instance, the kits for LXA4 and 15R-LXA4 show cross reactivity for resolvin D1 and 17R-resolvin D1, respectively, while the kit for LTB4 shows cross reactivity for both 5S-HETE and 5R-HETE (which could be formed during oxidative stress non-enzymatically as a racemic mixture). As such, since a more specific approach such as LC-MS/MS is not being used, the authors should at least more thoroughly establish the enzymatic origin of these mediators using additional distinct and selective inhibitors of 5-LOX in cells co-treated with ASA for 15R-LXA4 and inhibitors of FLAP, LTA4H, and a distinct inhibitor of 5-LOX (other than NDGA) for LTB4, as noted above. Response: We fully acknowledge that LC-MS/MS is the gold standard for accurate structural identification of lipids (and other biomolecules, for that matter, and that no antibody-based method is perfect). We provided our rationale for the use of ELISA rather than mass spectrometry in the responses listed above. In the revised manuscript, we have included the use of additional “specific” inhibitors of the LOX pathway (in the presence and absence of aspirin) to probe the LXA4 and 15R-LXA4 formation. Regarding the cross-reactivity of LXA4 and 15R-LXA4 ELISA kits with D-series resolvins, we observed NO evidence of resolving synthesis of the D-series in the absence of supplemental DHA. In the experiments described in the manuscript, we focused only on arachidonic acid-derived lipoxins (LXA4 and 15R-LXA4) using supplemental AA in the experiments. 

5. There are no figure legends are present in the manuscript and they should be included in a revision. Response: We apologize for the oversight in the original submission and have included a detailed figure legend in the revision.

---

## [Editor Report · Decision Letter 1]

8 Dec 2022

PONE-D-22-15963R1Aspirin modulates production of pro-inflammatory and pro-resolving mediators in endothelial cellsPLOS ONE

Dear Dr. Kniss,

Thank you for the response to the reviewers and mine comments. I believe most of the points were satisfactorily addressed. However, there a few things in the figures that need to be corrected:

Lanes in WB figures should be identified. One can assume that they are loaded in the same sequence of the densitometry graphs, but it would be better if they are labeled or aligned with graph bars.Figure legends, table S1, and densitometry graphs indicate GAPDH as the loading control, however all the WB figures show the loading control as beta-actin.Why there is no bar in "+ Zil" group in Fig 6B, +IL-1b panel? There was no detectable PGE2 produced or it was not done? If it was not done please indicate in the figura (and legend) with a ND. If there was no detectable PGE2 please mention in results.Please clarify in results section and legend how the experiments included in Fig. 6 were performed. Cells were washed after the initial 24-h incubation? They were washed after the incubation with AA ± inhibitors and before the A23187 chalenge? My guess would be yes and no, respectively.- It would probably make it easier for the reader if the bar colors are used in a standardized way: same color for the same lipid mediator; hashed bars (of the same color) for treatments; filled bars for stimulated cells, open bars for controls (with the outline indicating the color of the mediator analyzed) .Please submit your revised manuscript by Jan 22 2023 11:59PM. If you will need more time than this to complete your revisions, please reply to this message or contact the journal office at plosone@plos.org. Please include the following items when submitting your revised manuscript:A rebuttal letter that responds to each point raised by the academic editor and reviewer(s). You should upload this letter as a separate file labeled 'Response to Reviewers'.A marked-up copy of your manuscript that highlights changes made to the original version. You should upload this as a separate file labeled 'Revised Manuscript with Track Changes'.An unmarked version of your revised paper without tracked changes. You should upload this as a separate file labeled 'Manuscript'.If applicable, we recommend that you deposit your laboratory protocols in protocols.io to enhance the reproducibility of your results. Protocols.io assigns your protocol its own identifier (DOI) so that it can be cited independently in the future. For instructions see: https://journals.plos.org/plosone/s/submission-guidelines#loc-laboratory-protocols. Additionally, PLOS ONE offers an option for publishing peer-reviewed Lab Protocol articles, which describe protocols hosted on protocols.io. Read more information on sharing protocols at https://plos.org/protocols?utm_medium=editorial-email&utm_source=authorletters&utm_campaign=protocols.

We look forward to receiving your revised manuscript.

Kind regards,

Bruno Lourenco Diaz, Ph.D.

Academic Editor

PLOS ONE
---

## [Author Response · Author response to Decision Letter 1]

29 Dec 2022

Responses to Reviewers’ Comments:

General comments:

1. Concern about the specificity of ELISA assays to unequivocally determine the production of specific lipid mediators. Response: We recognize that the gold standard for the measurement of lipids in biological matrices is LC-MS/MS, typically using triple quadrupole mass spectrometry. In the current work, we used highly specific and thoroughly validated ELISAs from Cayman Chemical Co. to measure (6keto PGF1a, the stable metabolite of PGI2, PGE2, LTB4, LXA4 and 15-epi-LXA4 (Aspirin-triggered Lipoxin A4). Each of these solid-phase immunoassays has been used by many authors in high-profile publications. I have had in-depth discussions with the scientific staff at Cayman Chemical regarding any nuances with these assays. Each assay has been extensively tested for cross-reactivity (this information is included in the Supplementation Information included in the revised manuscript. We did not resort to LC-MS/MS in this particular manuscript due to financial limitations. Moreover, the use of 96-well ELISA formats permits the execution of large numbers of biological replicates that can be subjected to statistical testing. It would have been cost-prohibitive to have done the same number of replicates using LC-MS/MS. (My lab is working carefully with the Lipidomics Core facility at Wayne State University in Detroit, MI and with Cayman Chemical Co, Ann Arbor, MI to confirm the authenticity, accuracy, precision, specificity and sensitivity of the ELISA in comparison to conventional LC-MS/MS.). Dr. Gowdy has recently published a paper that demonstrated strong agreement between ELISA and LC-MS/MS values for PGE2 in lung tissue specimens (Yeager et al., Toxicol Sci, 2021; 183: 170, PMID: 34175951). Finally, we have conducted new experiments using pharmacological inhibitors of key enzymes and proteins in the lipid synthesizing pathways (e.g., 5-lipoxygenase inhibitor, Zileuton; FLAP inhibitor, MK-886) to confirm the results and conclusions in the original manuscript submission. 

2. 15-LO expression should be directed determined in HUVEC and preferably in some fashion that allow directly association of enzyme expression and the cell s containing endothelial markers such as immunofluorescence. Response: The culture system used in this work is HUVEC that are 100% endothelial in nature. There is no evidence of contamination from any other cell type. Thus, although it cannot be concluded that equal expression of all lipid synthesizing enzymes is found in each and every cell in the cultures, nonetheless, it is valid to conclude that whatever enzyme expression is detected (by qRT-PCR and western blotting) is the located in endothelial cells. 

Specific responses to reviewers’ comments:

1. Are the authors sure that the immunoassays applied to measure lipid mediators in complex mixtures of cell extract are specific for the indicated lipid mediator. They should verify their results with LC-MS analyses. Current LC-MS assays are more sensitive than the immunoassays. Response: 1. We are familiar with the attached paper outlining LC-MS/MS methods. We agree that LC-MS/MS continues to be the gold standard for lipidomics. In the current work we made the calculated decision to use ELISA methods in order to conduct many experiments using biological replicates. This would have been cost-prohibitive if LC-MS/MS were the selected method, since each sample costs approximately $100 to prepare and analyze. The ELISA assays used in this work were from Cayman Chemical Co. and have been extensively validated by the company and in-house for specificity and sensitivity. More detail on our in-house validation is presented in the Supplemental Information section of this revised manuscript. I have had extensive discussions with the scientific staff at Cayman Chemical and with Dr. Krishna Rao Maddipati, Director of the Lipidomics Core at Wayne State University regarding the pros and cons of ELISA versus LC-MS/MS. While LC-MS/MS is highly specific in identifying the precise structure of lipids, it is less sensitive than ELISA on a molar basis. Moreover, it is typically quite sensitive the matrix used. In the current work, we did not use cell extracts as implied by the reviewer’s question, rather we collected culture media. The medium was supplemented only with 0.5% charcoal-stripped FBS and endothelial cell growth supplement (from bovine brain extract). We tested the serum and supplement for the lipids in question and they were at or below the limit of detection. Thus, lipids measured in the media represent those produced by HUVEC and not contaminants from the supplements. Finally, the ELISAs for each lipid analyzed showed minimal cross-reactivity with related compounds at the physiological concentrations found in the media. We will continue to compare the ELISA assays with LC-MS/MS, but, in this work, we are confident in the methods of analysis. 

2. Each immunoassay has a measuring range and a LLOQ. Are all data points included in the manuscript for all lipid mediators within the measuring range of the respective immunoassay? Response: Indeed, all reported values for lipids in the manuscript exceed the LLOQ. We were very careful about this assertion. 

3. While expression analyses have been done for most enzymes necessary for lipid mediator production investigated in endothelial cells, the expression of ALOX15, necessary for LXA4 production is not shown in the manuscript. Response: In the revised manuscript, we have included qRT-PCR and western blots showing expression of 15-LOX-1, including the use of THP-1 and HL-60 cell extracts as positive controls. 

4. What levels of lipid mediators are present in the medium containing FCS used to incubate the HUVECs? Response: For all experiments, 0.5% charcoal-stripped FBS was used. We measured the levels of each lipid mediators in the 0.5% FCS and they are at or below the LLOQ for each ELISA. 

5. mPGES-1 protein expression is not visible on the western blot in Fig. 1H. Response: In the revised manuscript, we have included a better western blot image to illustrate the expression of mPGES-1, the enzyme that synthesizes PGE2. 

6. The results of the qPCR analysis are shown as fold increase only. However, it would be interesting to see how low the baseline levels for the different mRNAs are. Response: For the transcript expression experiments, we used the well-accepted 2-��Ct method of Schmittgen & Livak (Nature Protocols, 2008; 3: 1101). This method takes into account the change in target gene relative to the expression of a reference gene based upon the relative Ct values of each transcript. However, Table 1 in the original submission also includes the absolute Ct values for ALOX5, LTA4H and FLAP. In the revised manuscript, we also include the Ct values for the other transcripts were measured. 

7. There are no figure legends included in the manuscript, which makes the interpretation of the figures hard. Response: We sincerely apologize for this oversight. Somehow, the figure legend was not uploaded in the original submission. We have included the figure legend in the revised manuscript. 

8. It is not clear whether COX-1 and COX-2 are depicted in Figure 2A. Response: Apologies for the lack of clarity. The y-axis for COX-2 is on the left (with blue type) and the y-axis for COX-1 is on the right (with red type) of Figure 2A. We used color coding of COX-2 and COX-1 on the same graph to illustrate the inducibility of COX-2 and the lower level, constituent expression of COX-1. 

9. The western blot depicted in figures 4 A) and B) do not support the results depicted in the respective graphs. Response: We have prepared new western blots and densitometric analysis and have inserted these new data into Figure 4. 

10. Second last line on page 9 should be written Fig. 1C and not Fig. 1B. Response: We thank the reviewers for pointing out this error. We have made the correction in the text. 

11. Figure 8 is not readable due to a too low resolution of the TIF files. Response: These imaging glitches have been corrected. 

Reviewer #2: … a few grammar should be checked. Response: Done.

1. Please discuss the role of aspirin in low and high dose regarding anti-inflammatory and anti-thrombotic actions. We made a few comments about the use of low-dose aspirin for preeclampsia versus the of aspirin as anti-platelet therapy. 

2. In the results section pleased avoid to discuss your finding or to cite bibliography or to described the design of the study. Response: The only instance in which we cited previous publications in the Results section (see under IL-1b stimulates LTB4 synthesis in endothelial cells independent of leukocytes: Modulation by aspirin) is relevant and supports the conduct of the experiment and has been retained in the revised manuscript. 

3. The conclusion should be shorten. Response: Per the reviewer’s suggestion, we have shortened the Conclusions and Future Directions section. 

Reviewer #3: 

1. As the authors point out, 5-LOX is typically not highly expressed in endothelial cells, although some evidence suggests that it can be induced under certain conditions (e.g., cytomegalovirus infection, hypoxia, etc). Response: In the revised manuscript, we have included positive control THP-1 (monocyte/macrophage) and HL-60 (neutrophil) cell lysates for lipoxygenase western blots. 

2. The results show that native (15S) lipoxin A4 is produced in the HUVEC, impression of 15-LOX in this particular lot of cells (pooled donors I presume). Have the authors evaluated expression of 15-LOX type 1 or 2 in these cells? Response: In the revised manuscript, we include western blots of 15-LOX-1 and 15-LOX-2 expression in HUVEC. It is worthy noting that the HUVEC cultures were from pooled donors and several separate batches of cells were used in the experiments. 

3. The inhibitor used, NDGA can be non-specific, inhibit several lipoxygenases, and potentially anti-oxidant/free radical scavenging effects. Formation of non-enzymatic lipid oxidation products could occur in these cells under inflammatory conditions and structurally similar non-leukotrienes for example (see point 4 below). Thus, a more specific approach like siRNA (as described in Point 1) or a distinct inhibitor should additionally be used to definitively document the enzymatic origin of leukotrienes and lipoxins in these cells. Response: In the revised manuscript, we have included Zileuton, a specific 5-LOX inhibitor and MK-886, a FLAP inhibitor, in addition to NDGA, to document the specific LOX pathways involved in the production of leukotrienes and lipoxins in endothelial cells. 

4. (Related to Point 3): ELISAs for lipid mediators are known to have cross reactivity. For instance, the kits for LXA4 and 15R-LXA4 show cross reactivity for resolvin D1 and 17R-resolvin D1, respectively, while the kit for LTB4 shows cross reactivity for both 5S-HETE and 5R-HETE (which could be formed during oxidative stress non-enzymatically as a racemic mixture). As such, since a more specific approach such as LC-MS/MS is not being used, the authors should at least more thoroughly establish the enzymatic origin of these mediators using additional distinct and selective inhibitors of 5-LOX in cells co-treated with ASA for 15R-LXA4 and inhibitors of FLAP, LTA4H, and a distinct inhibitor of 5-LOX (other than NDGA) for LTB4, as noted above. Response: We fully acknowledge that LC-MS/MS is the gold standard for accurate structural identification of lipids (and other biomolecules, for that matter, and that no antibody-based method is perfect). We provided our rationale for the use of ELISA rather than mass spectrometry in the responses listed above. In the revised manuscript, we have included the use of additional “specific” inhibitors of the LOX pathway (in the presence and absence of aspirin) to probe the LXA4 and 15R-LXA4 formation. Regarding the cross-reactivity of LXA4 and 15R-LXA4 ELISA kits with D-series resolvins, we observed NO evidence of resolving synthesis of the D-series in the absence of supplemental DHA. In the experiments described in the manuscript, we focused only on arachidonic acid-derived lipoxins (LXA4 and 15R-LXA4) using supplemental AA in the experiments. 

5. There are no figure legends are present in the manuscript and they should be included in a revision. Response: We apologize for the oversight in the original submission and have included a detailed figure legend in the revision.

---

## [Editor Report · Decision Letter 2]

11 Jan 2023

PONE-D-22-15963R2Aspirin modulates production of pro-inflammatory and pro-resolving mediators in endothelial cellsPLOS ONE

Dear Dr. Kniss,Dear Dr. Kniss,

I believe there was some glitch in the system and your response does seem to match my comments on the revised version of the manuscript. I contacted the Journal Editorial office as soon as I received the new version, to try to identify the problem. Since I heven't heard from them and to avoid any further delays I transcribed my comments below:

Thank you for the response to the reviewers and mine comments. I believe most of the points were satisfactorily addressed. However, there a few things in the figures that need to be corrected:

Lanes in WB figures should be identified. One can assume that they are loaded in the same sequence of the densitometry graphs, but it would be better if they are labeled or aligned with graph bars.Figure legends, table S1, and densitometry graphs indicate GAPDH as the loading control, however all the WB figures show the loading control as beta-actin.Why there is no bar in "+ Zil" group in Fig 6B, +IL-1b panel? There was no detectable PGE2 produced or it was not done? If it was not done please indicate in the figura (and legend) with a ND. If there was no detectable PGE2 please mention in results.Please clarify in results section and legend how the experiments included in Fig. 6 were performed. Cells were washed after the initial 24-h incubation? They were washed after the incubation with AA ± inhibitors and before the A23187 chalenge? My guess would be yes and no, respectively.It would probably make it easier for the reader if the bar colors are used in a standardized way: same color for the same lipid mediator; hashed bars (of the same color) for treatments; filled bars for stimulated cells, open bars for controls (with the outline indicating the color of the mediator analyzed) .

We look forward to receiving your revised manuscript.

Kind regards,

Bruno Lourenco Diaz, Ph.D.

Academic Editor

PLOS ONE
---

## [Author Response · Author response to Decision Letter 2]

23 Feb 2023

Bruno Lourenco Diaz, PhD

Academic Editor, PLoS One

RE: Revised manuscript: PONE-D-22-15963

FEB 23 2023

Responses to Reviewers’ Comments:

General comments:

1. Concern about the specificity of ELISA assays to unequivocally determine the production of specific lipid mediators. Response: We recognize that the gold standard for the measurement of lipids in biological matrices is LC-MS/MS, typically using triple quadrupole mass spectrometry. In the current work, we used highly specific and thoroughly validated ELISAs from Cayman Chemical Co. to measure (6keto PGF1a, the stable metabolite of PGI2, PGE2, LTB4, LXA4 and 15-epi-LXA4 (Aspirin-triggered Lipoxin A4). Each of these solid-phase immunoassays has been used by many authors in high-profile publications. I have had in-depth discussions with the scientific staff at Cayman Chemical regarding any nuances with these assays. Each assay has been extensively tested for cross-reactivity (this information is included in the Supplementation Information included in the revised manuscript. We did not resort to LC-MS/MS in this particular manuscript due to financial limitations. Moreover, the use of 96-well ELISA formats permits the execution of large numbers of biological replicates that can be subjected to statistical testing. It would have been cost-prohibitive to have done the same number of replicates using LC-MS/MS. (My lab is working carefully with the Lipidomics Core facility at Wayne State University in Detroit, MI and with Cayman Chemical Co, Ann Arbor, MI to confirm the authenticity, accuracy, precision, specificity and sensitivity of the ELISA in comparison to conventional LC-MS/MS.). Dr. Gowdy has recently published a paper that demonstrated strong agreement between ELISA and LC-MS/MS values for PGE2 in lung tissue specimens (Yeager et al., Toxicol Sci, 2021; 183: 170, PMID: 34175951). Finally, we have conducted new experiments using pharmacological inhibitors of key enzymes and proteins in the lipid synthesizing pathways (e.g., 5-lipoxygenase inhibitor, Zileuton; FLAP inhibitor, MK-886) to confirm the results and conclusions in the original manuscript submission. 

2. 15-LO expression should be directed determined in HUVEC and preferably in some fashion that allow directly association of enzyme expression and the cell s containing endothelial markers such as immunofluorescence. Response: The culture system used in this work is HUVEC that are 100% endothelial in nature. There is no evidence of contamination from any other cell type. Thus, although it cannot be concluded that equal expression of all lipid synthesizing enzymes is found in each and every cell in the cultures, nonetheless, it is valid to conclude that whatever enzyme expression is detected (by qRT-PCR and western blotting) is the located in endothelial cells. 

Specific responses to reviewers’ comments:

1. Are the authors sure that the immunoassays applied to measure lipid mediators in complex mixtures of cell extract are specific for the indicated lipid mediator. They should verify their results with LC-MS analyses. Current LC-MS assays are more sensitive than the immunoassays. Response: 1. We are familiar with the attached paper outlining LC-MS/MS methods. We agree that LC-MS/MS continues to be the gold standard for lipidomics. In the current work we made the calculated decision to use ELISA methods in order to conduct many experiments using biological replicates. This would have been cost-prohibitive if LC-MS/MS were the selected method, since each sample costs approximately $100 to prepare and analyze. The ELISA assays used in this work were from Cayman Chemical Co. and have been extensively validated by the company and in-house for specificity and sensitivity. More detail on our in-house validation is presented in the Supplemental Information section of this revised manuscript. I have had extensive discussions with the scientific staff at Cayman Chemical and with Dr. Krishna Rao Maddipati, Director of the Lipidomics Core at Wayne State University regarding the pros and cons of ELISA versus LC-MS/MS. While LC-MS/MS is highly specific in identifying the precise structure of lipids, it is less sensitive than ELISA on a molar basis. Moreover, it is typically quite sensitive the matrix used. In the current work, we did not use cell extracts as implied by the reviewer’s question, rather we collected culture media. The medium was supplemented only with 0.5% charcoal-stripped FBS and endothelial cell growth supplement (from bovine brain extract). We tested the serum and supplement for the lipids in question and they were at or below the limit of detection. Thus, lipids measured in the media represent those produced by HUVEC and not contaminants from the supplements. Finally, the ELISAs for each lipid analyzed showed minimal cross-reactivity with related compounds at the physiological concentrations found in the media. We will continue to compare the ELISA assays with LC-MS/MS, but, in this work, we are confident in the methods of analysis. 

2. Each immunoassay has a measuring range and a LLOQ. Are all data points included in the manuscript for all lipid mediators within the measuring range of the respective immunoassay? Response: Indeed, all reported values for lipids in the manuscript exceed the LLOQ. We were very careful about this assertion. 

3. While expression analyses have been done for most enzymes necessary for lipid mediator production investigated in endothelial cells, the expression of ALOX15, necessary for LXA4 production is not shown in the manuscript. Response: In the revised manuscript, we have included qRT-PCR and western blots showing expression of 15-LOX-1, including the use of THP-1 and HL-60 cell extracts as positive controls. 

4. What levels of lipid mediators are present in the medium containing FCS used to incubate the HUVECs? Response: For all experiments, 0.5% charcoal-stripped FBS was used. We measured the levels of each lipid mediators in the 0.5% FCS and they are at or below the LLOQ for each ELISA. 

5. mPGES-1 protein expression is not visible on the western blot in Fig. 1H. Response: In the revised manuscript, we have included a better western blot image to illustrate the expression of mPGES-1, the enzyme that synthesizes PGE2. 

6. The results of the qPCR analysis are shown as fold increase only. However, it would be interesting to see how low the baseline levels for the different mRNAs are. Response: For the transcript expression experiments, we used the well-accepted 2-��Ct method of Schmittgen & Livak (Nature Protocols, 2008; 3: 1101). This method takes into account the change in target gene relative to the expression of a reference gene based upon the relative Ct values of each transcript. However, Table 1 in the original submission also includes the absolute Ct values for ALOX5, LTA4H and FLAP. In the revised manuscript, we also include the Ct values for the other transcripts were measured. 

7. There are no figure legends included in the manuscript, which makes the interpretation of the figures hard. Response: We sincerely apologize for this oversight. Somehow, the figure legend was not uploaded in the original submission. We have included the figure legend in the revised manuscript. 

8. It is not clear whether COX-1 and COX-2 are depicted in Figure 2A. Response: Apologies for the lack of clarity. The y-axis for COX-2 is on the left (with blue type) and the y-axis for COX-1 is on the right (with red type) of Figure 2A. We used color coding of COX-2 and COX-1 on the same graph to illustrate the inducibility of COX-2 and the lower level, constituent expression of COX-1. 

9. The western blot depicted in figures 4 A) and B) do not support the results depicted in the respective graphs. Response: We have prepared new western blots and densitometric analysis and have inserted these new data into Figure 4. 

10. Second last line on page 9 should be written Fig. 1C and not Fig. 1B. Response: We thank the reviewers for pointing out this error. We have made the correction in the text. 

11. Figure 8 is not readable due to a too low resolution of the TIF files. Response: These imaging glitches have been corrected. 

Reviewer #2: … a few grammar should be checked. Response: Done.

1. Please discuss the role of aspirin in low and high dose regarding anti-inflammatory and anti-thrombotic actions. We made a few comments about the use of low-dose aspirin for preeclampsia versus the of aspirin as anti-platelet therapy. 

2. In the results section pleased avoid to discuss your finding or to cite bibliography or to described the design of the study. Response: The only instance in which we cited previous publications in the Results section (see under IL-1b stimulates LTB4 synthesis in endothelial cells independent of leukocytes: Modulation by aspirin) is relevant and supports the conduct of the experiment and has been retained in the revised manuscript. 

3. The conclusion should be shorten. Response: Per the reviewer’s suggestion, we have shortened the Conclusions and Future Directions section. 

Reviewer #3: 

1. As the authors point out, 5-LOX is typically not highly expressed in endothelial cells, although some evidence suggests that it can be induced under certain conditions (e.g., cytomegalovirus infection, hypoxia, etc). Response: In the revised manuscript, we have included positive control THP-1 (monocyte/macrophage) and HL-60 (neutrophil) cell lysates for lipoxygenase western blots. 

2. The results show that native (15S) lipoxin A4 is produced in the HUVEC, impression of 15-LOX in this particular lot of cells (pooled donors I presume). Have the authors evaluated expression of 15-LOX type 1 or 2 in these cells? Response: In the revised manuscript, we include western blots of 15-LOX-1 and 15-LOX-2 expression in HUVEC. It is worthy noting that the HUVEC cultures were from pooled donors and several separate batches of cells were used in the experiments. 

3. The inhibitor used, NDGA can be non-specific, inhibit several lipoxygenases, and potentially anti-oxidant/free radical scavenging effects. Formation of non-enzymatic lipid oxidation products could occur in these cells under inflammatory conditions and structurally similar non-leukotrienes for example (see point 4 below). Thus, a more specific approach like siRNA (as described in Point 1) or a distinct inhibitor should additionally be used to definitively document the enzymatic origin of leukotrienes and lipoxins in these cells. Response: In the revised manuscript, we have included Zileuton, a specific 5-LOX inhibitor and MK-886, a FLAP inhibitor, in addition to NDGA, to document the specific LOX pathways involved in the production of leukotrienes and lipoxins in endothelial cells. 

4. (Related to Point 3): ELISAs for lipid mediators are known to have cross reactivity. For instance, the kits for LXA4 and 15R-LXA4 show cross reactivity for resolvin D1 and 17R-resolvin D1, respectively, while the kit for LTB4 shows cross reactivity for both 5S-HETE and 5R-HETE (which could be formed during oxidative stress non-enzymatically as a racemic mixture). As such, since a more specific approach such as LC-MS/MS is not being used, the authors should at least more thoroughly establish the enzymatic origin of these mediators using additional distinct and selective inhibitors of 5-LOX in cells co-treated with ASA for 15R-LXA4 and inhibitors of FLAP, LTA4H, and a distinct inhibitor of 5-LOX (other than NDGA) for LTB4, as noted above. Response: We fully acknowledge that LC-MS/MS is the gold standard for accurate structural identification of lipids (and other biomolecules, for that matter, and that no antibody-based method is perfect). We provided our rationale for the use of ELISA rather than mass spectrometry in the responses listed above. In the revised manuscript, we have included the use of additional “specific” inhibitors of the LOX pathway (in the presence and absence of aspirin) to probe the LXA4 and 15R-LXA4 formation. Regarding the cross-reactivity of LXA4 and 15R-LXA4 ELISA kits with D-series resolvins, we observed NO evidence of resolving synthesis of the D-series in the absence of supplemental DHA. In the experiments described in the manuscript, we focused only on arachidonic acid-derived lipoxins (LXA4 and 15R-LXA4) using supplemental AA in the experiments. 

5. There are no figure legends are present in the manuscript and they should be included in a revision. Response: We apologize for the oversight in the original submission and have included a detailed figure legend in the revision.

---

## [Editor Report · Decision Letter 3]

3 Mar 2023

Aspirin modulates production of pro-inflammatory and pro-resolving mediators in endothelial cells

PONE-D-22-15963R3

Dear Dr. Kniss,

We’re pleased to inform you that your manuscript has been judged scientifically suitable for publication and will be formally accepted for publication once it meets all outstanding technical requirements.

Kind regards,

Bruno Lourenco Diaz, Ph.D.

Academic Editor

PLOS ONE
---

## [Editor Report · Acceptance letter]

14 Apr 2023

PONE-D-22-15963R3 

ASPIRIN MODULATES PRODUCTION OF PRO-INFLAMMATORY AND PRO-RESOLVING MEDIATORS IN ENDOTHELIAL CELLS 

Dear Dr. Kniss:

I'm pleased to inform you that your manuscript has been deemed suitable for publication in PLOS ONE. Congratulations! Your manuscript is now with our production department. 

Kind regards, 

on behalf of

Dr. Bruno Lourenco Diaz 

Academic Editor

PLOS ONE